# Metal-mediated DNA strand displacement and molecular device operations based on base-pair switching of 5-hydroxyuracil nucleobases

Yusuke Takezawa [1] ✉, Keita Mori[1], Wei-En Huang[1], Kotaro Nishiyama[1], Tong Xing[1], Takahiro Nakama[1] & Mitsuhiko Shionoya [1] ✉

Rational design of self-assembled DNA nanostructures has become one of the fastest-growing research areas in molecular science. Particular attention is focused on the development of dynamic DNA nanodevices whose configuration and function are regulated by specific chemical inputs. Herein, we demonstrate the concept of metal-mediated base-pair switching to induce inter- and intramolecular DNA strand displacement in a metal-responsive manner. The 5-hydroxyuracil ($U^{OH}$) nucleobase is employed as a metal-responsive unit, forming both a hydrogen-bonded $U^{OH}$–A base pair and a metal-mediated $U^{OH}$–$Gd^{III}$–$U^{OH}$ base pair. Metal-mediated strand displacement reactions are demonstrated under isothermal conditions based on the base-pair switching between $U^{OH}$–A and $U^{OH}$–$Gd^{III}$–$U^{OH}$. Furthermore, metal-responsive DNA tweezers and allosteric DNAzymes are developed as typical models for DNA nanodevices simply by incorporating $U^{OH}$ bases into the sequence. The metal-mediated base-pair switching will become a versatile strategy for constructing stimuli-responsive DNA nanostructures, expanding the scope of dynamic DNA nanotechnology.

Rational design of self-assembled DNA nanostructures, termed DNA nanotechnology[1,2], is one of the fastest-growing research areas in molecular science. Hybridization based on strict base-paring rules has made DNA an excellent building block for constructing precisely defined molecular structures at the nanometer scale[3,4]. Of particular interest is the development of dynamic DNA nanodevices whose configuration and function are regulated by specific chemical inputs[5–7]. Such complex DNA systems are usually manipulated by toehold-mediated DNA strand displacement reactions (SDRs)[8,9], which use DNA or RNA strands as input signals. There is also growing interest in the operation of DNA nanodevices by diverse signals such as pH[10], light[11–13], and small molecules[14,15]. Precise chemical modifications[16] and conjugation with aptamers[17] and antibodies[18] have shown that DNA molecules can respond to specific chemical inputs or environmental stimuli.

Metal ions are often employed as chemical inputs in supramolecular chemistry[19]. Over the past few decades, a variety of metal-responsive molecular switches and machines have been synthesized[20–22] by exploiting the specific affinity between ligands and metal species. However, the construction of metal-responsive DNA systems is still in its infancy, as it has relied on limited types of metal–DNA interactions, such as potassium-dependent induction of G-quadruplexes[23] and metal-mediated formation of interstrand T–$Hg^{II}$–T and C–$Ag^{I}$–C complexes[24] (termed as metal-mediated base pairs[25–27]).

In this study, we demonstrate the concept of metal-mediated base-pair switching to induce inter- and intramolecular strand exchange in

[1]Department of Chemistry, Graduate School of Science, The University of Tokyo, 7-3-1 Hongo, Bunkyo-ku, Tokyo 113-0033, Japan.
✉e-mail: takezawa@chem.s.u-tokyo.ac.jp; shionoya@chem.s.u-tokyo.ac.jp

response to metal ions (Fig. 1). First, metal-mediated regulation of SDRs, one of the most fundamental processes for operating DNA nanodevices and computing circuits, is examined. Metal-mediated manipulation of DNA tweezers[28] and regulation of the catalytic activity of a deoxyribozyme (DNAzyme)[29] are also studied as typical models of DNA molecular devices. To enable rational sequence design of metal-responsive DNA structures, we utilize a noncanonical bifacial nucleobase, 5-hydroxyuracil ($U^{OH}$), which can form both a Watson−Crick type hydrogen-bonded $U^{OH}$−A base pair and a metal-mediated $U^{OH}$−M−$U^{OH}$ base pair (M = $Gd^{III}$, etc.) with its bidentate metal-binding site (i.e., an adjacent 4-carbonyl and a 5-hydroxy groups)[30–32]. Our previous studies have shown that the thermal stability of DNA duplexes containing $U^{OH}$−$U^{OH}$ mismatches is significantly enhanced by the formation of $U^{OH}$−$Gd^{III}$−$U^{OH}$ base pairs, while duplexes with $U^{OH}$−A base pairs are destabilized by the addition of $Gd^{III}$. This raises the possibility that the base-pairing partner of the $U^{OH}$ bases can be switched in response to $Gd^{III}$. We therefore conceived of the idea that $Gd^{III}$-mediated base-pair switching between $U^{OH}$−A and $U^{OH}$−$Gd^{III}$−$U^{OH}$ can be applied to the construction of dynamic DNA systems such as SDRs, DNA tweezers, and allosteric DNAzymes. Since $U^{OH}$ nucleobases can be easily synthesized and incorporated into DNA strands, sequence design strategies using metal-responsive $U^{OH}$ bases are expected to greatly expand the toolbox of dynamic DNA nanotechnology.

## Results

### Metal-mediated DNA strand displacement reactions (SDRs)

Dynamic DNA nanotechnology relies heavily on SDRs[8,9]. SDRs are usually initiated by the binding of an invading strand to a single-stranded segment called a toehold. The development of SDRs triggered by stimuli other than oligonucleotides is increasingly attracting attention as an expansion of the DNA nanotechnology toolbox. In the past, binding of small molecules to aptamers[15], pH changes[33], photo-irradiation[34,35], and metal−ligand interaction[36,37] have been employed to trigger SDRs. In this study, we investigated toehold-free, metal-mediated SDRs based on base-pair switching between $U^{OH}$−A and $U^{OH}$−$Gd^{III}$−$U^{OH}$ at the terminus of each strand.

Prior to the SDR experiments, metal-mediated regulation of DNA hybridization was first studied using oligonucleotides containing four $U^{OH}$ bases. Three 16-mer DNA strands were designed so that strand **1** can hybridize with strand **2** via $U^{OH}$−$Gd^{III}$−$U^{OH}$ base pairing and with

strand **3** via $U^{OH}$−A base pairing at their termini (Fig. 2a). The thermal stability of each duplex (i.e., **1·2** and **1·3**) in the absence and presence of $Gd^{III}$ ions was evaluated by melting analysis (Supplementary Figs. 1a and 2a). Under conditions without $Gd^{III}$ ions, the melting temperature ($T_m$) of duplex **1·3** containing four $U^{OH}$−A pairs was higher ($T_m = 45.3\,°C$) than that of duplex **1·2** containing four $U^{OH}$−$U^{OH}$ mismatch pairs (37.9 °C). Similar to previous studies[30,31], the addition of $Gd^{III}$ ions significantly increased the stability of duplex **1·2** ($\Delta T_m = +26.2\,°C$ with 4 equiv of $Gd^{III}$ ions). The stabilization of the duplex by $Gd^{III}$ was ascribed to the formation of interstrand $U^{OH}$−$Gd^{III}$−$U^{OH}$ complexes, which was confirmed by UV titration experiments and ESI-TOF mass spectrometry (Supplementary Fig. 1). On the other hand, duplex **1·3** was destabilized by the addition of $Gd^{III}$ ions ($\Delta T_m = -6.4\,°C$). This may be due to the weakening of the $U^{OH}$−A base pairing by the binding of $Gd^{III}$ ions to the $U^{OH}$ bases[30–32] (Supplementary Fig. 2). As a result, duplex **1·2** containing $U^{OH}$−$Gd^{III}$−$U^{OH}$ base pairs was found to be much more stable than duplex **1·3** containing $U^{OH}$−A base pairs upon addition of $Gd^{III}$ (Fig. 2b). These results indicate that the addition of $Gd^{III}$ ions reversed the order of the stability of the duplexes. The same trend was observed for DNA strands containing three $U^{OH}$ bases (Supplementary Fig. 3). Addition of $Gd^{III}$ ions (3 equiv) stabilized duplex **1′·2′** with three $U^{OH}$−$U^{OH}$ pairs ($\Delta T_m = +25.2\,°C$) and destabilized duplex **1′·3′** with three $U^{OH}$−A pairs ($\Delta T_m = -4.4\,°C$). The degree of duplex (de)stabilization ($\Delta T_m$) was slightly greater when four $U^{OH}$ bases were incorporated (Supplementary Fig. 4). Incorporation of more than four $U^{OH}$ bases was thought to cause undesirable intramolecular metal complexation. Therefore, DNA strands **1** and **2** containing four consecutive $U^{OH}$ bases were employed in the SDR experiments.

Next, an equimolar mixture of strands **1**, **2**, and **3** (labeled with a fluorophore FAM) was annealed in the absence and presence of $Gd^{III}$ ions to see which duplex (**1·2** or **1·3**) is preferentially formed. Native polyacrylamide gel electrophoresis (PAGE) analysis confirmed products containing strand **3** by visualization with FAM fluorescence (Fig. 2c) and all products by further staining (Supplementary Fig. 5). The results revealed that duplex **1·3** was mainly formed (81%) under the $Gd^{III}$-free condition. The addition of $Gd^{III}$ ions (4 equiv) resulted in the release of strand **3** (90%) and the formation of duplex **1·2**. UV spectroscopy showed that hybridization of duplex **1·2** was accompanied by metal complexation of the $U^{OH}$ bases (Fig. 2d). The yield of the formed duplex increased as the amount of $Gd^{III}$ ions increased from 0 to 4.0

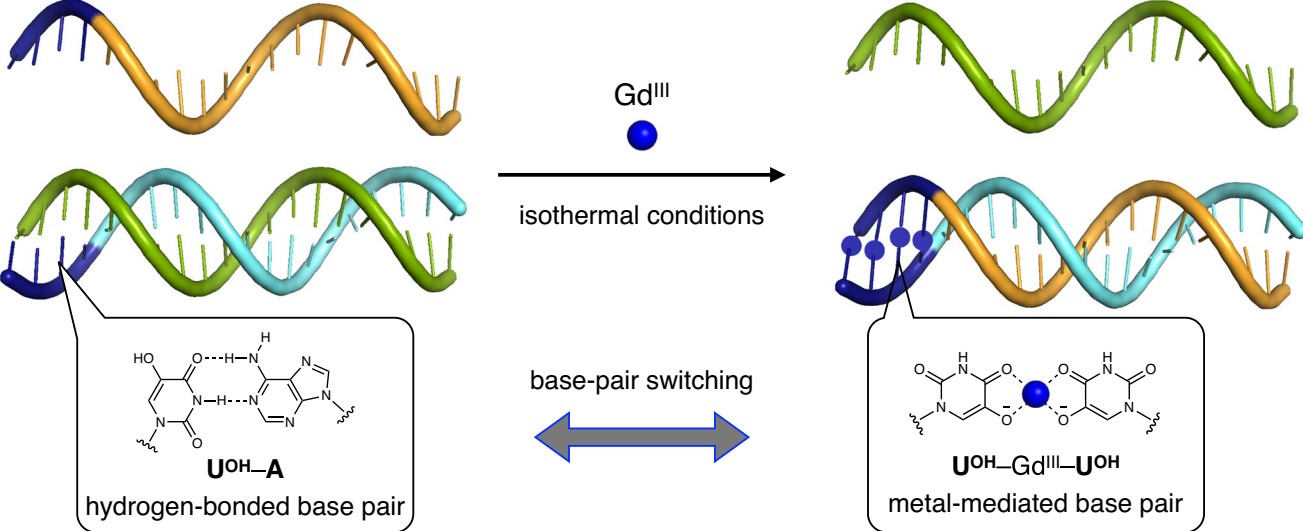

**Fig. 1 | Schematic representation of metal-mediated DNA strand displacement reactions (SDRs) induced by base-pair switching of 5-hydroxyuracil ($U^{OH}$) nucleobases.** Base-pair switching between hydrogen-bonded $U^{OH}$−A pairs and metal-mediated unnatural $U^{OH}$−$Gd^{III}$−$U^{OH}$ pairs can induce SDRs in response to $Gd^{III}$ ions. $U^{OH}$ nucleotides in the DNA strands are highlighted in navy blue. Other possible coordinating ligands on the $Gd^{III}$ ions, such as water molecules and neighboring nucleobases, are not shown for simplicity[30].

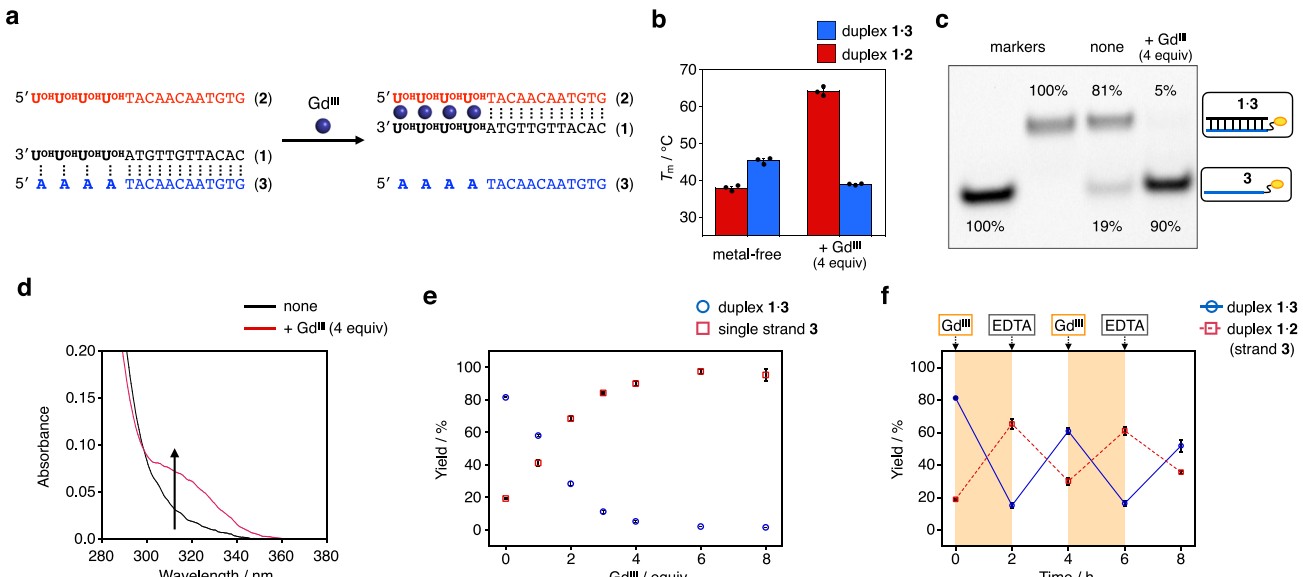

**Fig. 2 | Metal-dependent regulation of the hybridization behavior of a DNA strand containing U^OH bases. a** Schematic representation and base sequences. **b** Melting temperatures ($T_m$) of duplexes **1·2** (with $U^{OH}$–$U^{OH}$ base pairs) and **1·3** (with $U^{OH}$–A base pairs) in the absence and presence of Gd$^{III}$ ions (4 equiv). $N = 3$ independent experiments. Data are presented as mean values ± SEM. **c** Native PAGE analysis of an equimolar mixture of strands **1, 2**, and **3** in the absence and presence of Gd$^{III}$ ions (4 equiv). The samples were annealed prior to the analysis. The strand **3** was labeled with FAM for detection. The authentic samples (strand **3** and pre-annealed duplex **1·3**) were employed as the markers. Three independent experiments were performed. **d** UV absorption spectra of the mixture at 25 °C. $l = 1$ cm.

The samples were annealed after the addition of Gd$^{III}$ ions (4 equiv). **e** The yields of duplex **1·3** and single strand **3** in the presence of Gd$^{III}$ ions at different concentrations. The yields were estimated by comparing the band intensities on the gel with those of the markers. $N = 3$ independent experiments. Data are presented as mean values ± SEM. **f** Reversible regulation of the hybridization behaviors of U^OH-containing strand **1** by the alternate addition and removal of Gd$^{III}$ ions under isothermal conditions (25 °C). $N = 4$ independent experiments. Data are presented as mean values ± SEM. [DNA strand] = 2.0 µM each, [GdCl₃] = 0 or 8.0 µM (1 equiv per $U^{OH}$–$U^{OH}$ pair, otherwise noted) in 10 mM HEPES (pH 8.0), 100 mM NaCl. Source data are provided as a Source Data file.

equiv and was little changed by the addition of excess Gd$^{III}$ ions (Fig. 2e). These results show that hybridization preference of U^OH-containing strand **1** was changed in response to Gd$^{III}$.

Under isothermal conditions, sequential addition and removal of Gd$^{III}$ ions resulted in reversible and alternating formation of duplexes **1·2** and **1·3** (Fig. 2f). Four equivalents of Gd$^{III}$ ions and a chelating agent EDTA were added every 2 h at 25 °C. Native PAGE analysis of the hybridization products showed that the removal of Gd$^{III}$ ions by EDTA induced the formation of duplex **1·3** containing U^OH–A pairs. Subsequent addition of Gd$^{III}$ ions was found to regenerate duplex **1·2** via the interstrand U^OH–Gd$^{III}$–U^OH complexation. Thus, the hybridization partner of strand **1** containing U^OH bases was confirmed to be reversibly replaced by the addition and removal of Gd$^{III}$ ions.

Based on these results, metal-triggered SDRs were investigated by using longer DNA strands (Fig. 3a). Since the starting duplex (**4·6** or **7·9**) is long enough, the SDR should proceed via binding of the invading strand (**5** or **8**) and subsequent branch migration process to replace the incumbent strand (**6** or **9**). Like strands **1–3**, strands **4** and **5** have four consecutive U^OH bases, and strand **6** has four A bases at their termini. Strands **7, 8**, and **9** have an additional natural nucleotide (C or G) at the terminus to stabilize the duplexes. Even with the terminal G–C base pairs, the addition of Gd$^{III}$ reversed the thermal stability of the duplex containing U^OH–U^OH pairs and the duplex containing U^OH–A pairs (Supplementary Fig. 6). To examine metal-mediated SDRs, 4 equiv of Gd$^{III}$ ions were added to a mixture of a duplex containing U^OH–A base pairs (**4·6** or **7·9**) and a U^OH-modified invading strand (**5** or **8**, respectively). Two of the three strands (i.e., **4** and **5**, or **7** and **8**) were labeled with a fluorophore and a quencher, respectively. Thus, the progression of SDRs to form a duplex containing U^OH–Gd$^{III}$–U^OH base pairs (**4·5** or **7·8**) can be monitored by fluorescence quenching.

The results of the Gd$^{III}$-triggered SDRs are shown in Fig. 3b. The fluorescence decreased immediately after the addition of Gd$^{III}$ ions,

suggesting that SDR was initiated in response to Gd$^{III}$. The SDR proceeded at a similar rate regardless of the terminal G–C base pair. When the U^OH bases of strands **7** and **8** were replaced with natural T bases (**7t** and **8t**), little SDRs progressed with the addition of Gd$^{III}$ ions. This result confirms that the SDRs were triggered by metal complexation of the U^OH bases, inducing base-pair switching from U^OH–A to U^OH–Gd$^{III}$–U^OH. Under these conditions, Gd$^{III}$-mediated SDRs were slower than conventional toehold-mediated SDRs[8,9]. Complexation of U^OH–Gd$^{III}$–U^OH is completed within 1 min, as suggested by UV absorption analysis (Supplementary Fig. 7a). The results suggest that the subsequent branch migration is slowed down due to the structural distortion caused by U^OH–Gd$^{III}$–U^OH base pairing. In addition, the strand displacement may be delayed by the binding of the U^OH-modified invading strands (**5** or **8**) in a parallel orientation or by homo-dimerization of the invading strands via the U^OH–Gd$^{III}$–U^OH complexation (Supplementary Fig. 8). It is worth noting that the SDR using U^OH-containing strands is specifically triggered by certain lanthanide ions (Supplementary Fig. 9). The lanthanide ion Eu$^{III}$ triggered the SDR as Gd$^{III}$ does, while other transition metal ions such as Cu$^{II}$ and Zn$^{II}$ hardly induced SDR. The metal specificity is in good agreement with the fact that DNA duplexes containing U^OH–U^OH base pairs are stabilized only in the presence of lanthanide ions[30].

The addition of EDTA can remove Gd$^{III}$ ions from the U^OH–Gd$^{III}$–U^OH base pairs inside the duplexes, although the demetallation reaction is slower than complexation (Supplementary Fig. 7b). Under Gd$^{III}$-free conditions, the SDR can be initiated by adding strand **9** containing AAAAG at the terminus to duplex **7·8** with U^OH–U^OH mismatches (Supplementary Fig. 10). Thus, it was expected that the addition of EDTA would induce SDRs from duplexes with U^OH–Gd$^{III}$–U^OH (**4·5** or **7·8**) to duplexes with U^OH–A pairs (**4·6** or **7·9**, respectively). When EDTA ([EDTA]/[Gd$^{III}$] = 1.0) was added to a mixture of a Gd$^{III}$-bridged duplex (**4·5** or **7·8**) and the other strand (**6** or **9**), the fluorescence was gradually

**a**

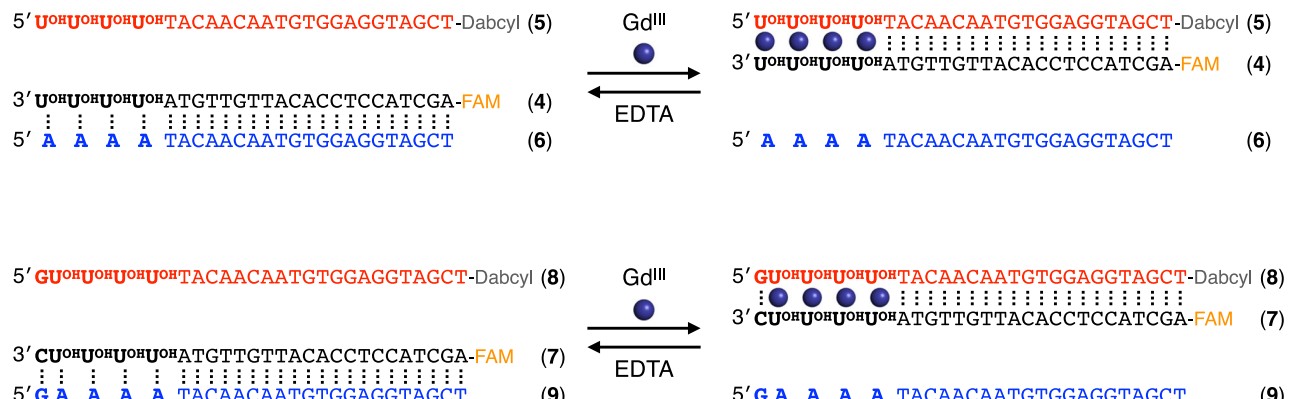

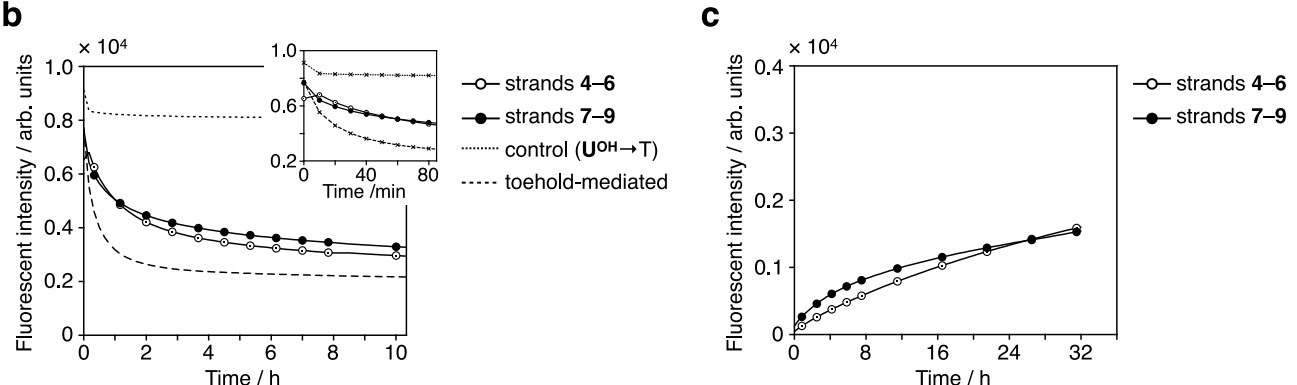

**b**

**c**

Fig. 3 | **Metal-mediated DNA strand displacement reactions (SDRs) based on the base-pair switching of U^OH bases. a** Design of metal-mediated SDRs. Strands **4** and **7** were labeled with a fluorophore (FAM) and strands **5** and **8** with a quencher (Dabcyl). **b** Time-course analysis of Gd^III-triggered SDRs. The reaction was started by the addition of Gd^III ions (4 equiv). In the control experiment, strands **7t** and **8t**, in which U^OH bases are replaced with natural T bases, were used instead of strands **7** and **8**. The toehold-mediated reaction was carried out with FAM-labeled **7t, 8 s** (21-mer that lacks the terminal 5′-GU^OHU^OHU^OHU^OH-3′), and Dabcyl-labeled **9**. **c** Time-course analysis of EDTA-triggered SDRs. The reaction was started by the addition of EDTA (4 equiv) to remove Gd^III ions. [DNA strand] = 2.0 μM each in 10 mM HEPES buffer (pH 8.0), 100 mM NaCl, 25 °C. The SDRs were monitored by the changes in the fluorescence of the FAM. $\lambda_{ex}$ = 495 nm, $\lambda_{em}$ = 519 nm. Source data are provided as a Source Data file.

increased. This result indicates that SDRs proceeded in the reverse direction (Fig. 3c). The slow rate of the EDTA-induced SDRs could be improved by increasing the number of U^OH bases or by adding terminal bases only to the toehold and the incoming strand.

As discussed above, Gd^III-triggered SDRs using the U^OH base as a metal-binding site are now possible. Gd^III destabilized the U^OH–A base pairs and induced the formation of the U^OH–Gd^III–U^OH base pairs. This base-pair switching promoted the displacement of the A-containing incumbent strands with the U^OH-containing invading strand. The SDRs were found to proceed in the reverse direction upon removal of the Gd^III ions, which induces base-pair switching from U^OH–Gd^III–U^OH to U^OH–A. Note that the SDRs developed in this study are reversible even though there are no exposed toehold regions in the strands. Since Gd^III ions are unlikely to interact with natural nucleobases, the Gd^III-triggered SDRs using unnatural U^OH nucleobases would have significant advantages, especially when integrated into cascade SDRs such as DNA computing circuits. Thus, SDRs based on the base-pair switching between U^OH–A and U^OH–Gd^III–U^OH may provide a useful strategy for designing a wide variety of stimuli-responsive DNA molecular systems.

**Development of metal-responsive DNA tweezers using U^OH bases**
Based on the results of the strand displacement experiments, we further applied the U^OH bases to the development of metal-responsive

DNA molecular devices. As the simplest model, a pair of DNA tweezers[28,38–40] was created that switch its structure upon metal complexation of the U^OH bases (Fig. 4a). DNA tweezers are nanodevices consisting of two arms that take two forms, closed and open. Despite their simple structure, DNA tweezers have been applied to capture target proteins[41] and regulate enzyme functions[42], making them an important prototype of DNA molecular machines. The base sequence of the metal-responsive tweezers was designed based on Yurke's original DNA tweezers[28] operated by toehold-mediated SDR using fuel strands (Fig. 4b). The tweezers consist of two DNA duplex arms connected by a single-strand hinge. Hybridization of complementary oligonucleotides (strand **d**) at both dangling ends locks the DNA tweezers into a closed state. The U^OH bases were introduced into strand **d** so that the tweezers are closed by U^OH–A base pairing. The addition of Gd^III ions was expected to destabilize the U^OH–A base pairs and to induce the release of strand **d** associated with the intramolecular U^OH–Gd^III–U^OH base pairing, which would open the tweezers.

The structure of the U^OH-modified DNA tweezers was first evaluated by native PAGE analysis (Fig. 4c). Each band was characterized by comparing with the mobility of reference samples in which U^OH bases were replaced with T bases. When all strands were annealed in the absence of Gd^III ions, a band corresponding to the closed state was mainly observed. In the presence of Gd^III ions, a new band with higher

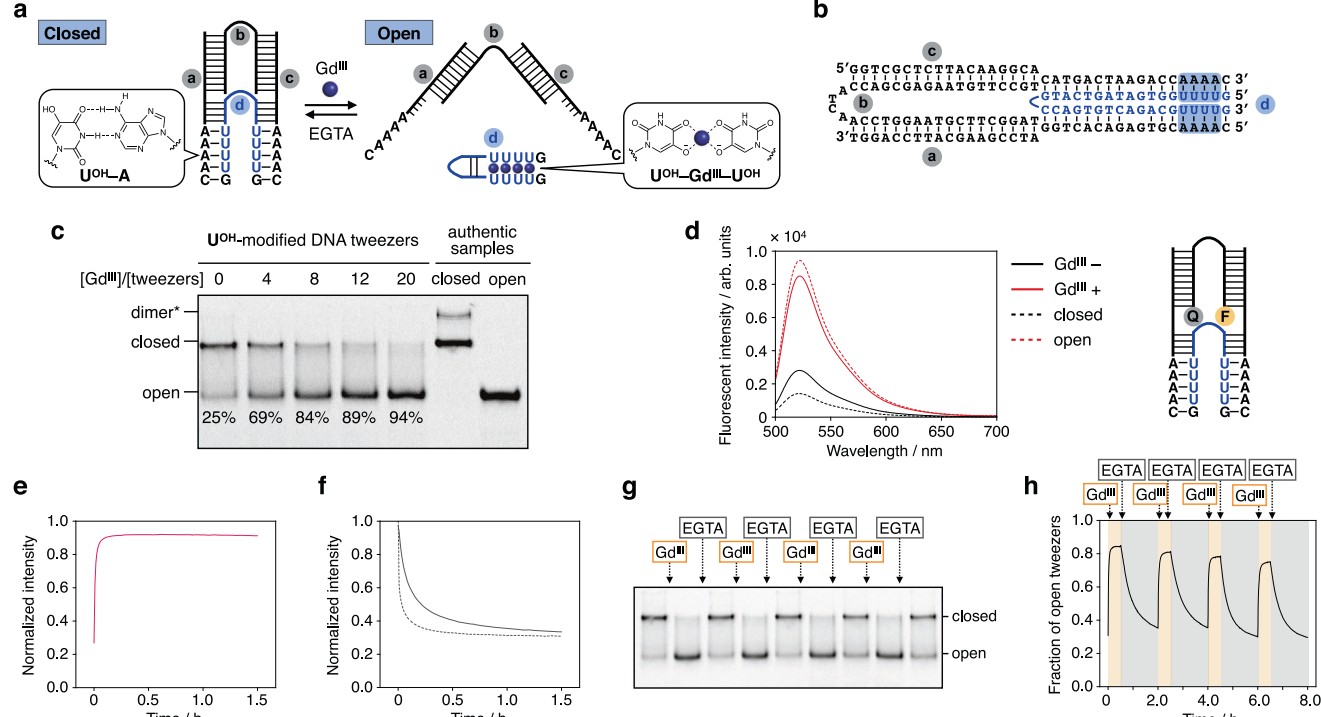

**Fig. 4 | Metal-mediated operation of DNA tweezers with U^OH bases. a** Design of Gd^III-responsive DNA tweezers. U represents **U^OH** nucleotides. **b** The base sequence of the **U^OH**-modified DNA tweezers used in this study. **c** Native PAGE analysis of the structures of DNA tweezers in the absence and presence of Gd^III ions. Strand **b** was labeled with FAM for detection. Two authentic samples, the DNA tweezers without strand **d** (open state) and with a closing strand containing T in place of **U^OH** (closed state), were used as the markers. 15% gel at 20 °C. Three independent experiments were performed. (*) A dimeric structure[28] in which two DNA tweezers are connected by binding of strand **d**. **d** Fluorescence analysis of the tweezers. Strand **b** was labeled with FAM and Dabcyl at both termini. $\lambda_{ex}$ = 495 nm, 25 °C. The samples were annealed prior to the measurement. **e** Opening of the tweezers triggered by the Gd^III addition. $\lambda_{ex}$ = 495 nm, $\lambda_{em}$ = 519 nm, 25 °C. **f** Closure of the tweezers triggered by the addition of EGTA (ethyleneglycol bis(2-aminoethyl ether)-*N,N,N',N'*-tetraacetic acid), which selectively removes Gd^III ions in the buffer containing Mg^II ions. Closure caused by the addition of strand **d** is also shown as a broken line. **g, h** Repeated opening and closing of the tweezers by alternating addition and removal of Gd^III ions observed by native PAGE. Three independent experiments were performed. The bands were assigned by comparison with the authentic samples. **h** Repeated opening and closing of the tweezers observed by fluorescence analysis. [DNA] = 2.0 μM each in 10 mM HEPES buffer (pH 8.0), 100 mM NaCl, 3 mM MgCl₂, [GdCl₃] = 0 or 24 μM (otherwise noted), [EGTA] = 0 or 24 μM. Source data are provided as a Source Data file.

mobility appeared, indicating the formation of an open state consisting of strands **a**, **b**, and **c** only. The open state was formed more efficiently by increasing the amount of Gd^III ions. In the following experiments, the DNA tweezers were operated with 12 equiv of Gd^III ions.

The metal-mediated switching between the closed and open states was further examined by FRET analysis (Fig. 4d). The hinge strand **b** was labeled with a fluorophore and a quencher. The fluorescence observed in the presence of Gd^III ions (red solid line) was significantly higher than that observed in the absence of Gd^III ions (black solid line). This suggests that the closed state was formed in the absence of Gd^III ions, in which the FRET pairs were in close proximity to each other, while the open state was formed upon metal addition. The yields of both states were approximated by comparison with the control experiments (broken lines). The results showed that 73 ± 4% of the tweezers were closed in the absence of Gd^III ions, whereas 94 ± 1% were open after annealing with 12 equiv of Gd^III ions. Both values were in good agreement with the yields estimated from the band intensities in the PAGE analysis (Fig. 4c).

Next, the time course of the metal-mediated tweezer action was analyzed at a constant temperature of 25 °C. The fluorescence increased rapidly with the addition of Gd^III ions, indicating that the metal addition triggered the tweezer opening under isothermal conditions (Fig. 4e). The opening reaction was faster than the SDR progression, reaching equilibrium within 5 min. This may be because the structure transformation of the DNA tweezers is based

on the intramolecular **U^OH**–Gd^III–**U^OH** base pairing and the strand **d** is sufficiently short to dissociate easily. Conversely, the addition of a chelating agent EGTA (ethyleneglycol bis(2-aminoethyl ether)-*N,N,N',N'*-tetraacetic acid), which can selectively remove Gd^III ions in the buffer containing Mg^II, caused the tweezers to close (Fig. 4f). The closing reaction was nearly complete in 1 h, but the rate of closure was slower than that induced by the addition of strand **d** (broken line). This may be due to the somewhat slower rate at which Gd^III ions are removed from the **U^OH**–Gd^III–**U^OH** base pairs inside the duplexes (Supplementary Fig. 7b). These results indicate that the **U^OH**-modified DNA tweezers can be operated reversibly in response to Gd^III ions.

Furthermore, the successive opening and closing of the tweezers were demonstrated by the sequential addition and removal of Gd^III ions. Gd^III ions (12 equiv) and EGTA (12 equiv) were added alternately to the pre-annealed DNA tweezers at a time interval of 2 h at 25 °C. Native PAGE and FRET analyses showed that the DNA tweezers were opened and closed repeatedly, undergoing structural changes four times (Fig. 4g, h). Although Gd^III–EGTA complexes accumulated in this reaction, metal complexation proved to be useful in manipulating DNA tweezers under isothermal conditions. All results shown here suggest that the tweezers operate based on metal-mediated base-pair switching of the **U^OH** bases without any toehold regions. Thus, the **U^OH** nucleobase can be used as a versatile building block for constructing metal-responsive DNA nanodevices other than molecular tweezers.

## Development of metal-dependent allosteric DNAzymes using $U^{OH}$ bases

Deoxyribozymes (DNAzymes) are DNA oligomers with catalytic activity that were originally obtained by an in vitro selection method (SELEX) using nucleic acid libraries with a random sequence[29]. Among a variety of DNAzymes, RNA-cleaving DNAzymes, which cleave oligonucleotide substrates containing a ribonucleotide, are widely used to construct DNA molecular machines and computing circuits[43,44]. Allosteric regulation of DNAzyme activity is of growing interest as a promising strategy for constructing stimuli-responsive DNA systems[45]. We have previously synthesized $Cu^{II}$-responsive allosteric DNAzymes by introducing metal-mediated artificial base pairs ($H$–$Cu^{II}$–$H$ or $Im^C$–$Cu^{II}$–$Im^C$) that stabilize DNA duplexes only in the presence of specific metal ions[46–50]. Here, we developed a metal-responsive allosteric DNAzyme by applying a base-pair switching system between hydrogen-bonded $U^{OH}$–A base pairs and metal-mediated $U^{OH}$–$Gd^{III}$–$U^{OH}$ base pairs (Fig. 5a).

The metal-responsive DNAzyme was designed based on the secondary structure of the RNA-cleaving NaA43 DNAzyme reported previously[51] (Fig. 5b). Three $U^{OH}$–$U^{OH}$ base pairs were incorporated into the stem duplex as metal-binding sites. The sequence of the surrounding bases was modified to form a catalytically inactive structure with $U^{OH}$–A base pairs in the absence of $Gd^{III}$ ions. The addition of $Gd^{III}$ ions was expected to switch the base pairing from $U^{OH}$–A to $U^{OH}$–$Gd^{III}$–$U^{OH}$, resulting in a structure transformation from the inactive to the catalytically active form.

The RNA-cleaving activity of the $U^{OH}$-modified DNAzyme (named as $U^{OH}$-DNAzyme) was evaluated in the absence and presence of $Gd^{III}$ ions. The reaction was initiated by the addition of a fluorophore-labeled substrate (10 equiv), and the time course of the reaction was monitored by denaturing PAGE (Fig. 5c). In the absence of $Gd^{III}$ ions, the activity of $U^{OH}$-DNAzyme was efficiently suppressed ($k_{obs} = 4.8 \times 10^{-3}$ h$^{-1}$) compared to that of the original NaA43 DNAzyme ($4.2 \times 10^{-1}$ h$^{-1}$). This result indicates that the $U^{OH}$-DNAzyme adopted a

catalytically inactive structure in the absence of $Gd^{III}$ ions. The addition of 3 equiv of $Gd^{III}$ ions enhanced the catalytic reaction of $U^{OH}$-DNAzyme ($k_{obs} = 6.9 \times 10^{-2}$ h$^{-1}$). The activity of the unmodified DNAzyme was also increased by the addition of $Gd^{III}$ ions ($k_{Gd+}/k_{Gd-} = 1.9$), but the $Gd^{III}$-induced activity enhancement of the $U^{OH}$-DNAzyme was much more pronounced ($k_{Gd+}/k_{Gd-} = 14.4$). These results show that the interactions between $U^{OH}$ and $Gd^{III}$ play a pivotal role in DNAzyme activation. In control experiments using T-DNAzyme, in which all $U^{OH}$ bases were replaced with thymine (T) bases, no substrate cleavage was observed in the presence of $Gd^{III}$ ions, suggesting that $U^{OH}$-DNAzyme was activated by the complexation of $U^{OH}$ bases with $Gd^{III}$ ions. UV spectroscopy also revealed the appearance of a new absorption band around 310 nm by the addition of $Gd^{III}$ ions (Supplementary Fig. 11), suggesting metal complexation of the $U^{OH}$ bases[30,31]. It can be concluded that the activity of $U^{OH}$-DNAzyme is allosterically enhanced by the formation of $U^{OH}$–$Gd^{III}$–$U^{OH}$ base pairs, stabilizing the catalytically active structure. It should also be noted that the $U^{OH}$-DNAzyme showed more efficient switching capability ($k_{Gd+}/k_{Gd-} = 14.4$) than the $Cu^{II}$-responsive DNAzyme ($k_{Cu+}/k_{Cu-} = 5.9$)[47] developed by incorporating an $H$–$Cu^{II}$–$H$ base pair into the same parent DNAzyme. This is thought to be due to a unique design strategy based on the bifacial nature of the $U^{OH}$ bases, whereby the inactive secondary structure (with $U^{OH}$–A base pairs) and the active structure (with $U^{OH}$–$Gd^{III}$–$U^{OH}$ pairs) are formed stably in the absence and presence of $Gd^{III}$ ions, respectively.

We found that the DNAzyme activity during the reaction can be controlled by adding and removing $Gd^{III}$ ions. The substrate was treated with $U^{OH}$-DNAzyme in the absence of $Gd^{III}$ ions for 2 h, and then $Gd^{III}$ ions were added. Time-course analysis showed that the addition of $Gd^{III}$ ions increased the DNAzyme activity to the level observed with $Gd^{III}$ ions from the beginning (Fig. 5d). When the chelating agent EDTA was added to remove $Gd^{III}$ ions, $U^{OH}$-DNAzyme was immediately inactivated (Fig. 5e). It is interesting to note that the DNAzyme activity changed a short time after the addition or removal of $Gd^{III}$ ions. This is probably because the intramolecular structure transformation requires a little time.

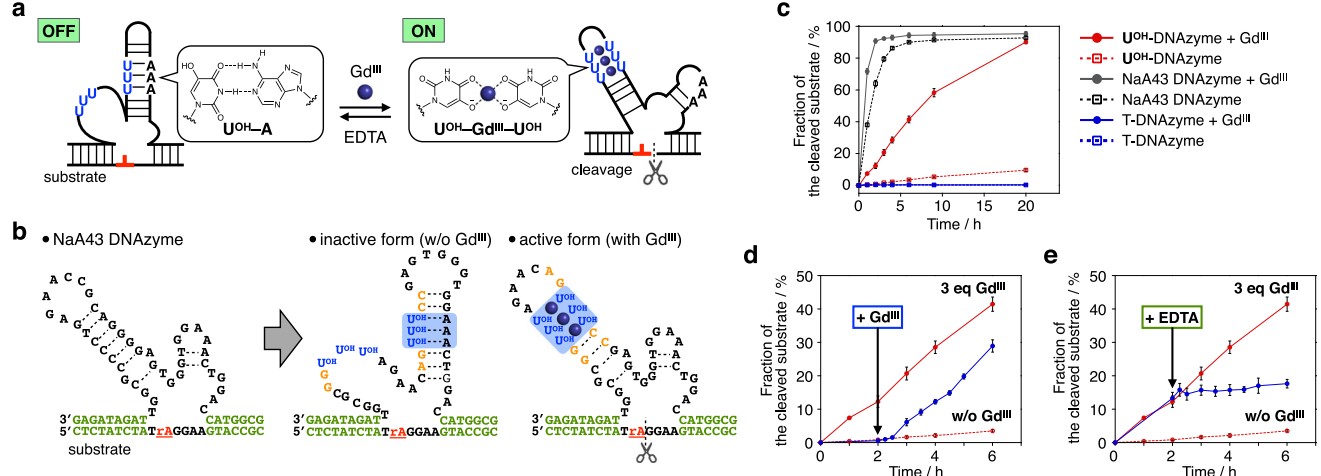

**Fig. 5 | Development of a $Gd^{III}$-responsive allosteric DNAzyme based on the base-pair switching of $U^{OH}$ bases. a** Schematic representation of the $Gd^{III}$-responsive allosteric DNAzyme. U represents $U^{OH}$ nucleotides. **b** Base sequence of the $U^{OH}$-modified DNAzyme ($U^{OH}$-DNAzyme). DNA strands containing an adenosine ribonucleotide (rA) at the cleavage site were used as the substrate. Both the catalytically inactive form (without $Gd^{III}$) and the active form (with $Gd^{III}$) are shown. The original NaA43 DNAzyme was modified by introducing $U^{OH}$ bases, and surrounding nucleobases (shown in orange) were further altered to stabilize the inactive structure in the absence of $Gd^{III}$ ions. **c** RNA-cleaving activity of $U^{OH}$-DNAzyme, the original NaA43 DNAzyme, and T-DNAzyme in the absence and presence of $Gd^{III}$ ions (3 equiv). $N = 3$ (for $U^{OH}$-DNAzyme) and $N = 4$ (for the others) independent experiments. Data are presented as mean values ± SEM. T-DNAzyme has T bases in place

of $U^{OH}$ bases, thus lacking the metal binding affinity. The substrate cleavage was analyzed by denaturing PAGE using a substrate labeled with a fluorophore (FAM). **d** Activation of $U^{OH}$-DNAzyme by the addition of $Gd^{III}$ ions (3 equiv). The activity of $U^{OH}$-DNAzyme in the presence and absence of $Gd^{III}$ ions (3 equiv) is also shown by the red lines. $N = 3$ independent experiments. Data are presented as mean values ± SEM. **e** Deactivation of $U^{OH}$-DNAzyme by removing $Gd^{III}$ ions. An equimolar amount of EDTA was added to remove the $Gd^{III}$ ions. $N = 4$ independent experiments. Data are presented as mean values ± SEM. [$U^{OH}$-DNAzyme] = 1.0 μM, [substrate] = 10 μM, [$GdCl_3$] = 3.0 μM (3 equiv, where applicable), [EDTA] = 3.0 μM (3 equiv, where applicable) in 10 mM HEPES (pH 7.0), 100 mM NaCl, 25 °C. Source data are provided as a Source Data file.

All results presented here show that the activity of $U^{OH}$-DNAzyme is allosterically regulated in response to $Gd^{III}$ ions under isothermal conditions. This is due to molecular design based on the base-pair switching between $U^{OH}$–A and $U^{OH}$–$Gd^{III}$–$U^{OH}$. Although the duplex structure around the $U^{OH}$–$Gd^{III}$–$U^{OH}$ base pairs is presumed to be distorted[30], the $U^{OH}$-modified DNAzyme showed catalytic activity in the presence of $Gd^{III}$ ions. Note that $U^{OH}$ bases served as metal-responsive switches to induce DNA structural conversion even when located at internal positions. Sequence design based on the base-pair switching of $U^{OH}$ may be applicable not only to allosteric DNAzymes but also to the development of various metal-responsive functional DNA devices.

## Discussion

In this study, we have shown that $Gd^{III}$-dependent DNA nanostructure transformation is possible by base-pair switching between hydrogen-bonded $U^{OH}$–A base pairs and metal-mediated $U^{OH}$–$Gd^{III}$–$U^{OH}$ base pairs. In the absence of $Gd^{III}$ ions, $U^{OH}$ forms $U^{OH}$–A base pairs, which are similar to Watson–Crick T–A pairs. When $Gd^{III}$ ions are added as external stimuli, the $U^{OH}$–A base pair is destabilized and the $U^{OH}$–$Gd^{III}$–$U^{OH}$ base pairs can be preferentially formed instead. The addition and removal of $Gd^{III}$ ions result in the exchange of hybridization partners of the $U^{OH}$-containing strands even under isothermal conditions. The introduction of multiple $U^{OH}$ bases at the termini of DNA strands enabled a $Gd^{III}$-triggered strand displacement reaction (SDR), although it is not as fast as standard toehold-mediated SDRs. This metal-responsive SDR is reversible and does not require any toehold sequences, making it a versatile approach to creating dynamic DNA molecular devices.

The $Gd^{III}$-mediated base-pair switching was further applied to the operation of DNA tweezers as a simple model for DNA molecular devices. Gel electrophoresis and FRET analysis confirmed that the DNA tweezers repeatedly open and close in response to $Gd^{III}$ ions. We also confirmed that this method can be applied to control the activity of a catalytic DNA (DNAzyme). Here, we designed a $Gd^{III}$-dependent transformation between a catalytically inactive structure containing $U^{OH}$–A base pairs and an active structure with $U^{OH}$–$Gd^{III}$–$U^{OH}$ base pairs. As a result, we succeeded in activating and inactivating the DNAzyme in response to the metal. Thus, $U^{OH}$ nucleobases are expected to be widely applied to manipulate DNA molecular devices and machines using metal ions as external stimuli.

The use of metal ions as external stimuli in dynamic DNA nanotechnology has the following characteristics. (1) Because coordination bonds are generally more stable than hydrogen bonds, metal-mediated base pairing can significantly alter the stability of DNA duplexes[25]. This is advantageous for inducing strand displacement and structure transformation of DNA constructs. (2) Metal coordination is fundamentally reversible, allowing repeated manipulation of metal-DNA systems, as demonstrated with the DNA tweezers in this study. Reversible SDRs could be useful in developing advanced DNA computing circuits. (3) A wide variety of metal ions are available for this purpose, and equimolar amounts or small excesses of metal ions are sufficient as stimuli when suitable ligands are used. Multiple metal species can also be used as orthogonal stimuli based on specific interactions between metal ions and ligands[46,48]. (4) SDRs triggered by biologically relevant metal ions have a wide range of biological applications.

Metal-mediated base pairs composed of natural nucleobases (i.e., T–$Hg^{II}$–T and C–$Ag^{I}$–C)[52–55] have been used to construct metal-responsive DNA devices. However, these metal ions can bind to T and C bases at undesired positions that are difficult to predict, making sequence design not easy. We have worked to rationally design metal-responsive DNA supramolecules by introducing metal-mediated base pairs with unnatural ligand-type nucleobases[46–50]. Compared to these conventional approaches, metal-mediated base-pair switching dramatically alters duplex stability and hybridization behavior, making it

suitable for dynamic control of DNA nanostructures. Moreover, because only modified nucleotides were used as additional building blocks, the strategies presented here are highly compatible with standard DNA nanotechnologies.

The basic behaviors seen in the metal-mediated SDRs and DNA nanodevice presented here are based on the unique concept of metal-mediated base-pair switching of modified pyrimidine bases. In conventional pH-responsive[10] and light-responsive systems[11–13], only (de)hybridization of DNA strands is controlled by external stimuli. In contrast, our approach can reversibly switch two different states: a structure with $U^{OH}$–A base pairs (i.e., inactive state) and one with $U^{OH}$–$Gd^{III}$–$U^{OH}$ base pairs (active state), as clearly demonstrated using the $U^{OH}$-DNAzyme. These features provide a useful basis for the development of DNA nanodevices, molecular machines, computing circuits, and more advanced systems[56]. In contrast to the selection methods by which various metal-dependent DNAzymes have been found[57], our strategy is primarily focused on the rational design of metal-responsive DNA materials. Modification of existing functional DNAs is also a suitable approach, as the DNA sequences can be easily designed by replacing natural T bases with $U^{OH}$ bases. $U^{OH}$ nucleoside and $U^{OH}$-modified oligonucleotides are readily available, and $U^{OH}$ bases function as metal-triggered switches not only at the termini of the DNA strands but also internally. Therefore, the $U^{OH}$ bases will be a useful building block for designing metal-responsive dynamic DNA architectures.

It should also be mentioned that metal-mediated SDR at lower concentrations is required for practical application in DNA nanotechnology. Because of the potential interaction of $Gd^{III}$ ions with phosphate groups, the use of other metal ions that rarely interact with natural DNA may be more appropriate for applications in complex systems consisting of many DNA strands. Metal-mediated base-pair switching can also be possible with other 5-substituted pyrimidine bases such as 5-carboxyuracil (caU)[58] and N,N-dicarboxymethyl-5-aminouracil (dcaU)[59], both of which were found to form more stabilizing metal-mediated base pairs (caU–$Cu^{II}$–caU and dcaU–$Gd^{III}$–dcaU) as well as hydrogen-bonded base pairs (caU–A and dcaU–A). We believe that this method can be extended to other metal ions by designing bifacial nucleobases with different coordination functionality at the 5-position or by modifying cytosine bases. Thus, the principle of metal-mediated base-pair switching appears to be a versatile approach that extends the scope of dynamic DNA nanotechnology. The application of these bifacial nucleobases to the metal-dependent reconfiguration of DNA nanostructures is currently under investigation in our laboratory.

## Methods

### Materials and equipment

All natural DNA strands, including FAM-labeled strands, Dabcyl-labeled strands, and substrate strands containing a riboadenosine (rA) were purchased from Japan Bio Service Co., Ltd. (Saitama, Japan) at HPLC purification grade. The sequences of the DNA strands used in this study are summarized in Supplementary Table 1. Oligonucleotide concentrations were determined based on UV absorbance at 260 nm. $GdCl_3$·$6H_2O$ (99.9% purity) purchased from Soekawa Chemical Co. was used as a metal source without further purification. The gels were analyzed using Gel Doc EZ Imager and Image Lab software ver. 6.1.0 (Bio-Rad). All the graphs were produced by KaleidaGrapgh ver. 5.0.3 (HULINKS).

### DNA synthesis

DNA strands containing 5-hydroxyuracil ($U^{OH}$) nucleobases were synthesized on an automated DNA synthesizer (NTS M-2-MX DNA/RNA synthesizer)[30]. The DNA synthesis was carried out on a 1-μmol scale in a DMTr-on mode with ultramild deprotection phosphoramidites and reagents (Glen Research). Note that the phosphoramidite monomer of $U^{OH}$ deoxynucleoside was synthesized following the literature[60,61] or

purchased from Glen Research (catalog No. 10-1053). The coupling time of **U^OH** was extended to 15 min. The products were deprotected using 28% NH₃ aqueous solution at room temperature for 2–3 h and then purified and detritylated using a PolyPak II cartridge (Glen Research). Some of the **U^OH**-containing strands were prepared by the ligation of a shorter **U^OH**-containing strand and a natural strand by using a T4 DNA ligase. The oligonucleotides were purified by reverse-phase HPLC (Waters XBridge C18 column) or by denaturing poly-acrylamide gel electrophoresis. All DNA strands were identified by MALDI-TOF or ESI-TOF mass spectrometry (Supplementary Table 2).

### Duplex melting analysis

The samples were prepared by mixing the DNA strands (2 μM) in 10 mM HEPES buffer (pH 8.0) containing 100 mM NaCl. After the addition of Gd^III ions (0–6 equiv), the solutions were heated to 85 °C and cooled slowly to 4 °C at a rate of −1.0 °C/min. Absorbance at 260 nm was monitored by UV-1800 and UV-1900 spectro-photometers (Shimadzu) equipped with a TMSPC-8 temperature controller while the temperature was raised from 4 to 85 °C at a rate of 0.2 °C/min. Normalized absorbance shown in the figures was cal-culated as follows:

$$\text{Normalized } \Delta A_{260} = \{A_{260}(t\,°C) - A_{260}(4\,°C)\} / \{A_{260}(85\,°C) - A_{260}(4\,°C)\}. \quad (1)$$

The melting temperature ($T_m$) was determined as the inflection point of the melting curve using LabSolutions $T_m$ analysis software ver. 1.40 (Shimadzu) with a 17-point adaptive smoothing program. The average $T_m$ values of at least three independent runs were calculated.

### Mass spectrometry of the DNA duplexes

Electrospray ionization-time-of-flight (ESI-TOF) mass spectra were recorded on a Waters Micromass LCT premier. The samples (40 μM) were prepared in 20 mM NH₄OAc buffer (pH 7.0) and annealed just before the measurements (from 85 to 4 °C, −1.0 °C/min).

### Metal-dependent regulation of the hybridization behavior of U^OH-containing DNA strands

Strand **3** was labeled with FAM. The samples were prepared by mixing the DNA strands (2 μM each) in 10 mM HEPES buffer (pH 8.0) con-taining 100 mM NaCl. The solutions were annealed (85 °C → 4 °C, −1.0 °C/min) in the absence or presence of Gd^III ions. For the reversible regulation shown in Fig. 2f, after the solutions were annealed (85 °C → 25 °C, −1.0 °C/min), GdCl₃ (4 equiv) and EDTA (4 equiv) were alternately added every 2 h at 25 °C. Native gel electrophoresis was carried out with 20% polyacrylamide gel in a cool incubator (4 °C). The bands were detected by FAM fluorescence, and the yield of each product was calculated by comparing the band intensities of the authentic samples on the same gel. The averages of at least three runs are summarized. The gel shown in Supplementary Fig. 5 was stained with SYBR Gold. The UV spectra of the samples were also recorded at 5 °C on a JASCO V-730 spectrometer.

### Metal-mediated DNA strand displacement reactions (SDRs)

Strands **4** and **7** were labeled with a fluorophore (FAM) and strands **5** and **8** with a quencher (Dabcyl). The initial duplexes (**4·6** or **7·9** for Gd^III-triggered reaction, and **4·5** or **7·8** for EDTA-triggered reaction) were prepared by combining the two strands (2.0 μM each) in 10 mM HEPES buffer (pH 8.0) containing 100 mM NaCl in the absence or presence of Gd^III ions (4 equiv). After annealing from 85 to 25 °C (−1.0 °C/min), the mixture was kept at 25 °C. After the addition of the other strand, the SDR was initiated by the addition of Gd^III ions (4 equiv) or EDTA (4 equiv). The reaction progress was monitored by the changes in the fluorescence intensity of the FAM ($\lambda_{ex}$ = 495 nm, $\lambda_{em}$ = 519 nm) using an FP-8350 fluorometer (JASCO). The fluorescent intensity was recorded in 10-min intervals for 8 h and subsequently in 30-min intervals. In the control experiments, the toehold-mediated SDR was started by adding the invading strands to the pre-annealed duplex. In the metal selectivity test (Supplementary Fig. 9), the fluorescence intensity ($\lambda_{ex}$ = 495 nm, $\lambda_{em}$ = 520 nm) was recorded on a Varioskan LUX plate-reader (Thermo Fisher).

### Metal-mediated operation of the DNA tweezers

Strand **b** was labeled with FAM at the 5′ end for native PAGE analysis or with FAM and Dabcyl at both termini for FRET analysis. The samples were prepared by combining DNA strands **a, b, c**, and **d** (2 μM each) in 10 mM HEPES buffer (pH 8.0) containing 100 mM NaCl and 3 mM MgCl₂. The solutions were annealed (85 °C → 25 °C, −1.0 °C/min) in the absence or presence of Gd^III ions (4, 8, 12, and 20 equiv). Native PAGE was carried out with 15% polyacrylamide gel in an incubator (20 °C). The bands were detected by FAM fluorescence, and the yields of the closed and the open states were estimated by comparing the band intensities. The averages of at least three runs are summarized. A mixture of strands **a, b**, and **c** was used as a marker corresponding to the open state. A mixture of strands **a, b, c**, and **d′**, in which **U^OH** nucleotides were replaced with T, was used to indicate the closed state. For the FRET analysis, the fluorescence spectra were measured at 25 °C ($\lambda_{ex}$ = 495 nm).

### Time-course study of the operation of the DNA tweezers

Strand **b** was labeled with FAM and Dabcyl at both termini. DNA strands **a, b, c**, and **d** (2 μM each) were mixed in 10 mM HEPES buffer (pH 8.0) containing 100 mM NaCl and 3 mM MgCl₂. The mixtures were annealed (85 °C → 25 °C, −1.0 °C/min) in the absence or presence of Gd^III ions (12 equiv). Gd^III ions (12 equiv) or EGTA (12 equiv) were added while maintaining 25 °C. The change in the fluorescence intensity was monitored in 30-s intervals at 25 °C ($\lambda_{ex}$ = 495 nm, $\lambda_{em}$ = 519 nm). In the control experiments, strand **d** was added to the pre-annealed mixture of strands **a, b**, and **c**.

### Repetitive opening and closing of the tweezers

The sample was prepared in the absence of Gd^III ions, as described above. With keeping the temperature at 25 °C, Gd^III ions (12 equiv) or EGTA (12 equiv) were alternately added at the defined time points. The reaction progress was monitored by native PAGE and FRET analysis.

### Metal-mediated regulation of DNAzyme activity

A DNAzyme strand (1.0 μM) was annealed (85 °C → 25 °C, −1.0 °C/min) in a reaction buffer (10 mM HEPES (pH 7.0), 100 mM NaCl) in the presence or absence of GdCl₃ (3 equiv). The RNA-cleaving reaction was initiated by adding a FAM-labeled substrate (10 equiv). After incubat-ing at 25 °C, the reaction was stopped by adding a 3:1 mixture of 7 M urea and the loading buffer. The progress of the DNAzyme reaction was analyzed by denaturing PAGE. The percentage of the cleaved product ($F$) was calculated as follows:

$$F(\%) = I_c / (I_c + I_u) \times 100, \quad (2)$$

where $I_c$ and $I_u$ are the band intensities of the cleaved and the uncleaved substrate, respectively. The apparent first-order rate constants ($k_{obs}$) were calculated from the initial velocity determined from the time points when $F$ was less than 20%.

### Regulation of DNAzyme activity during the reaction

For the Gd^III-triggered DNAzyme activation, the RNA-cleaving reaction was initiated in the absence of Gd^III ions. After 2 h, GdCl₃ (3 equiv) was added. For the deactivation by the removal of Gd^III ions, the reaction was started in the presence of Gd^III ions (3 equiv). After 2 h, a chelating agent EDTA (3 equiv) was added to remove Gd^III ions. The reaction progress was monitored by denaturing PAGE as described above.

**Reporting summary**

Further information on research design is available in the Nature Portfolio Reporting Summary linked to this article.

## Data availability

All the data supporting the results of this study are available in this paper and the Supplementary Information. Source data are provided with this paper.

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

## Acknowledgements
This work was supported by JSPS KAKENHI Grant Numbers JP18H02081, JP21H02055, JP22K19100 to Y.T., JP21J11325 (Grant-in-Aid for JSPS Fellows) to K.M., and JP21H05022 to M.S. and MEXT KAKENHI Grant Numbers JP21H00384 (Molecular Engine), JP21H05866, JP23H04399 (Molecular Cybernetics) to Y.T., and JP16H06509 (Coordination Asymmetry) to M.S., Japan.

## Author contributions
Y.T., K.M., and M.S. conceived and designed the project. K.M., W.-E.H., K.N., and T.X. carried out the experiments with assistance from Y.T. and T.N. Y.T., K.M., and W.-E. H. analyzed the data. Y.T., K.M., and M.S. wrote the manuscript.

## Competing interests
The authors declare no competing interests.
