## [Peer Review File · Nature Communications]

Metal-mediated DNA strand displacement and molecular device operations based on base-pair switching of 5-hydroxyuracil nucleobasesEditorial Note: Parts of this Peer Review File have been redacted as indicated to maintain the confidentiality of unpublished data in Comment 4 on page 12.

REVIEWER COMMENTS

Reviewer #1 (Remarks to the Author):

This is an innovative paper showing how metal ions can mediate base pair switching when using hydroxyuracil-A base pairs. In this way, metal ions such as Gd³⁺ can induce strand exchange. The authors investigate DNA strand displacement using Gd³⁺ as the trigger.

In figure 2 the authors show how Gd³⁺ can bind to the DNA pair and the reversibility with EDTA. Fig 3 shows the strand displacement. Fig 4 shows the application of tweezers using this system, and Fig 5 shows the application of an allosteric DNAzyme using this system.

A general comment relates to the specificity of Gd³⁺ in this switching mechanism. How does this compare to other similarly sized transition metal ions? This would be important to investigate considering the applications highlighted in the discussion. A broader set of controls with the close related ions would be important.

Some other more specific comments:

1. The evidence in Fig 2 would be far stronger if there were variants with 2 bases and 6 bases capable to bind the Gd³⁺ and then showing that the molar titrations still hold in these circumstances, proving a single Gd³⁺ binds per base pair.
2. Fig 4 any reason the switch from EDTA to EGTA for reversibility - would it be better have consistency across figures?
3. Fig. 5 titrate with Gd³⁺ to prove there are three equivalents of Gd³⁺.

Reviewer #2 (Remarks to the Author):

In their manuscript, Takezawa, Shionoya et al. report a new strategy for the design of stimuli-responsive DNA-based molecular devices. Strand-displacement reactions (SDR) are fundamental reactions in dynamic DNA nanotechnology. They typically require at least one toehold-containing DNA strand. The authors now report a toehold-free SDR by using the artificial nucleobase 5-hydroxyuracil (UOH). This nucleobase forms a regular hydrogen-bonded base pair with adenine, but can also form a metal-mediated homo base pair. Towards this end, the authors make use of Gd(III) ions.

They convincingly show in competition experiments that a single-stranded DNA oligonucleotide containing consecutive UOH bases prefers to form a duplex with complementary adenine residues in the absence of Gd(III), whereas a duplex with complementary UOH residues preferentially forms in the presence of Gd(III). The hybridization behavior can be switched repeatedly by the addition and removal (via chelators) of Gd(III), respectively. Hence, SDRs are possible with internal UOH residues without the requirement for a toehold.

Next, the authors show the general applicability of their new strategy by implementing it into two well-established DNA nanodevices, namely DNA tweezers and a DNAzyme. In both cases, the toehold-free metal-induced switching is very well compatible with the function of the DNA nanodevices.

All conclusions drawn in the manuscript are the result of several complementary experimental setups, using e.g. UV spectroscopy, DNA melting analysis, native PAGE, and fluorescence spectroscopy. Time-dependent measurements allow first conclusions to be

drawn with respect to the kinetics of the strand-displacement reactions.

In summary, the authors present a highly innovative and important contribution to the field of dynamic DNA nanotechnology. The use of Gd(III)-mediated base pairs certainly expands the scope of DNA nanotechnology. The manuscript is of interest to readers from various chemical disciplines. I consider it highly important and recommend its acceptance after minor revisions as follows.

- 1) The authors should briefly state why they chose to use four consecutive UOH residues in all their oligonucleotides. Why not fewer (or more)?
- 2) The mass spectrum shown in Figure S1c shows peaks for duplexes bearing three and four Gd(III) ions with similar intensities. Even though this is not relevant for the SDRs, it would be interesting to know whether indeed all four binding sites are occupied by Gd(III) or whether one regularly remains unoccupied. Do the authors have any indication in this respect?
- 3) The difference between Figures 2c and S3 should be stated in the text (presumably visualization via FAM fluorescence to show only constructs containing oligonucleotide 3 and via SYBR gold staining to show all constructs?). Moreover, the identity of the “markers” should be stated in the figure legend. Are these molecular weight markers?
- 4) Figure S4 reports data for shortened oligonucleotides 7', 8', and 9'. What was the reason for not using full-length oligonucleotides 7, 8, and 9?
- 5) Figure 3b could be improved by including an inset with an expanded section for the first hour of reaction.
- 6) Line 144: The authors state that “The slow rate of the EDTA-induced SDRs could be improved by increasing the number of UOH bases or by adding terminal bases that facilitate the binding of the incoming strand.” The latter statement is not in agreement with the Figure 3c, which shows that the presence of one terminal base in fact reduced the rate.
- 7) Figure 4c shows a band referred to as “dimer*^o”. This dimer is neither explained in the text nor in the figure legend.

Reviewer #3 (Remarks to the Author):

This is an interesting and exhaustive study on DNA nanodevices based on the ability of 5-hydroxyuracil to form a canonical Watson—Crick base pair with adenine as well as a Gd(III)-mediated homo base pair. The manuscript is very well written, the experiments elegantly designed and carefully executed and the conclusions fully consistent with the data obtained. However, the authors should be more forthcoming regarding comparison of the present study with previous ones from the same group. For example, the U^{OH}-Gd(III)-U^{OH} base pair has been reported previously (references 28—30) and the conceptual novelty of the metal-dependent allosteric DNAzymes is also questionable in light of references 46—49. If the DNAzyme presented in this paper shows some improvement over those described in references 46—49, it should be explained more clearly. I leave it to the Editor to decide whether the novelty of the results merits publication in Nature Communications or whether a less demanding journal, such as Scientific Reports, would be more appropriate.

Reviewer #4 (Remarks to the Author):

The idea of metal ion mediated strand displacement to control nucleic acid molecular devices is interesting and the authors have demonstrated the working principles of this method. Overall it is a good study. Here are a few questions that need to be addressed before its publication:

- 1) How versatile this method is, can this be extended to a wide range of metal ions. Why do

people want to use these metals to drive the strand displacement, what is the biological function if any. Isothermal DSD is definitely interesting.

2) in Figure 3 in the gel showing the reversible switching, there appears to be less than 100% of switching, what is causing this? Will this negatively impact any downstream DSD if more complicated devices are designed?

3)The system can be more systematic by including studies of length/sequence/structure dependence. It would improve the work by include the use of metal mediated strand displacement in reconfiguration of DNA nanostructure (1D/2D systems).

A List of Corrections for NCOMMS-22-43185

We have revised our manuscript based on the reviewers' comments and suggestions as follows. The changes we have made are also highlighted in yellow in the manuscript.

Reviewer #1

Comment 1. This is an innovative paper showing how metal ions can mediate base pair switching when using hydroxyuracil-A base pairs. In this way, metal ions such as Gd^{3+} can induce strand exchange. The authors investigate DNA strand displacement using Gd^{3+} as the trigger. In figure 2 the authors show how Gd^{3+} can bind to the DNA pair and the reversibility with EDTA. Fig 3 shows the strand displacement. Fig 4 shows the application of tweezers using this system, and Fig 5 shows the application of an allosteric DNAzyme using this system. A general comment relates to the specificity of Gd^{3+} in this switching mechanism. How does this compare to other similarly sized transition metal ions? This would be important to investigate considering the applications highlighted in the discussion. A broader set of controls with the close related ions would be important.

Answer 1. We sincerely thank this reviewer for the positive assessment and important suggestions. According to the reviewer's suggestion, we have additionally performed the strand displacement reactions (SDRs) with various metal ions using a plate reader. It was also found that the lanthanide ion Eu^{III} causes SDR, as Gd^{III} does, while other transition metal ions such as Cu^{II} and Zn^{II} hardly induce SDR. These results confirmed the specificity of SDR using U^{OH} -containing strands for lanthanide ions. The metal specificity of SDR is in good agreement with the fact that DNA duplexes containing $U^{OH}-U^{OH}$ base pairs are thermally stabilized only in the presence of certain lanthanide ions (Ref. 28). The additional experimental results are shown in Supplementary Fig. S9 and the discussion is added to the main text as follows.

Supplementary Fig. S9 | Time course analysis of the SDRs triggered by adding various metal ions. [DNA strand] = 2.0 μ M each in 10 mM HEPES buffer (pH 8.0), 100 mM NaCl, 25 °C. The SDRs were monitored with a plate reader by the changes in the fluorescence of FAM (λ_{ex} = 495 nm, λ_{em} = 520 nm). The invader strand 5 was added

to the pre-annealed duplex **4·6**, and then the metal ions (4 equiv.) were added to initiate the reaction.”

“In addition, the strand displacement may be delayed by the binding of the U^{OH} -modified invading strands (**5** or **8**) in a parallel orientation or by homo-dimerization of the invading strands via the $\text{U}^{\text{OH}}\text{-Gd}^{\text{III}}\text{-U}^{\text{OH}}$ complexation (Supplementary Fig. S8). It is worth noting that the SDR using U^{OH} -containing strands is specifically triggered by certain lanthanide ions (Supplementary Fig. S9). The lanthanide ion Eu^{III} triggered the SDR as Gd^{III} does, while other transition metal ions such as Cu^{II} and Zn^{II} hardly induced SDR. The metal specificity is in good agreement with the fact that DNA duplexes containing $\text{U}^{\text{OH}}\text{-U}^{\text{OH}}$ base pairs are stabilized only in the presence of lanthanide ions²⁸.”

“Methods

Metal-mediated DNA strand displacement reactions (SDRs). (snip) The reaction progress was monitored by the changes in the fluorescence intensity of the FAM ($\lambda_{\text{ex}} = 495 \text{ nm}$, $\lambda_{\text{em}} = 519 \text{ nm}$) using an FP-8350 fluorometer (JASCO). The fluorescent intensity was recorded in 10-minute intervals for 8 h and subsequently in 30-minute intervals. In the control experiments, the toehold-mediated SDR was started by adding the invading strands to the pre-annealed duplex. In the metal selectivity test (Supplementary Fig. S9), the fluorescence intensity ($\lambda_{\text{ex}} = 495 \text{ nm}$, $\lambda_{\text{em}} = 520 \text{ nm}$) was recorded on a Varioskan LUX plate-reader (Thermo Fisher).”

Comment 2. The evidence in Fig 2 would be far stronger if there were variants with 2 bases and 6 bases capable to bind the Gd^{3+} and then showing that the molar titrations still hold in these circumstances, proving a single Gd^{3+} binds per base pair.

Answer 2. We thank the reviewer for the comments. The stoichiometry was supported by the UV titration experiments shown in Supplementary Fig. S1b. The spectra varied almost linearly in the range $[\text{Gd}^{\text{III}}]/[\text{duplex}] = 0$ to 4 and remained almost unchanged in the presence of excess Gd^{III} , confirming that all U^{OH} bases bind to the Gd^{III} ions. In the mass spectrum (Fig. S1c), the duplex with four Gd^{III} ions ($1\cdot 2\cdot \text{Gd}^{\text{III}}_4$) was mainly observed, further confirming the binding of four Gd^{III} ions to the duplex containing four $\text{U}^{\text{OH}}\text{-U}^{\text{OH}}$ base pairs (i.e., one Gd^{III} ion for one $\text{U}^{\text{OH}}\text{-U}^{\text{OH}}$ base pair). Note that the duplex with three Gd^{III} ions ($1\cdot 2\cdot \text{Gd}^{\text{III}}_3$) was also observed, but this may due to the dissociation of the outermost Gd^{III} ions during the measurements. Similar results have already been reported for DNA duplexes containing three $\text{U}^{\text{OH}}\text{-U}^{\text{OH}}$ base pairs in a different sequence (Ref. 28). Since the structural analysis of the metallo-DNA duplexes is ongoing, more detailed discussion of the coordination geometry of the $\text{U}^{\text{OH}}\text{-Gd}^{\text{III}}\text{-U}^{\text{OH}}$ base pairs will be reported elsewhere. These explanations are added to the caption of Supplementary Fig. S1.

“**Supplementary Fig. S1 | Gd^{III}-mediated stabilization of DNA duplex containing U^{OH}-U^{OH} base pairs.** (snip) **b** UV absorption spectra of duplex 1·2 in the presence of different concentrations of Gd^{III} ions. [Gd^{III}]/[duplex] = 0, 1, 2, 3, 4 (solid lines), 5, and 6 (broken lines). In 10 mM HEPES buffer (pH 8.0), 100 mM NaCl. 5 °C, *l* = 1 cm. The spectra varied almost linearly in the range [Gd^{III}]/[duplex] = 0 to 4 and remained almost unchanged in the presence of excess Gd^{III} ions, suggesting that four Gd^{III} ions bind to the four U^{OH}-U^{OH} base pairs. **c** ESI-TOF mass spectrum of duplex 1·2 in the presence of Gd^{III} ions (4 equiv). [duplex] = 40 μM, [Gd^{III}]/[duplex] = 4.0 in 20 mM NH₄OAc buffer (pH 7.0). Negative mode. The existence of the trinuclear complex (1·2·Gd^{III}₃) is likely due to the dissociation of the outermost Gd^{III} ion during the measurement.”

Comment 3. Fig 4 any reason the switch from EDTA to EGTA for reversibility - would it be better have consistency across figures?

Answer 3. We thank the reviewer for the important comment. The DNA tweezers were operated in a buffer containing 3 mM Mg^{II} ions. EDTA cannot be used to remove Gd^{III} ions under the condition because of its high affinity for Mg^{II} ions. Therefore, EGTA, which has lower affinity for Mg^{II} and binds strongly to Gd^{III}, was employed to manipulate the DNA tweezers. According to the reviewer’s comment, the figure legend was revised as follows.

“**Fig. 4 | Metal-mediated operation of DNA tweezers with U^{OH} bases.** ... **f** Closure of the tweezers triggered by the addition of EGTA (ethyleneglycol bis(2-aminoethyl ether)-*N,N,N',N'*-tetraacetic acid), which selectively removes Gd^{III} ions in the buffer containing Mg^{II} ions.”

Comment 4. Fig. 5 titrate with Gd³⁺ to prove there are three equivalents of Gd³⁺.

Answer 4. We thank the reviewer for the important comment. Preliminary experiments showed that the addition of excess Gd^{III} ions slightly increased the DNAzyme activity. We think that titration experiments cannot elucidate the stoichiometry because (i) the activation of the U^{OH}-DNAzyme is caused by both Gd^{III}-mediated destabilization of the U^{OH}-A pairs and Gd^{III}-mediated formation of the U^{OH}-Gd^{III}-U^{OH} base pairs, and (ii) the activity of the parent NaA43 DNAzyme was slightly enhanced by the addition of Gd^{III} ions ($k_{Gd^+}/k_{Gd^-} = 1.9$). From the UV spectral changes shown in Fig. S11 ($\Delta\epsilon_{310} = 1.5 \times 10^4 \text{ L} \cdot \text{mol}^{-1} \cdot \text{cm}^{-1}$), it was roughly estimated that more than 70% of the U^{OH} bases formed the U^{OH}-Gd^{III}-U^{OH} base pairs ($\Delta\epsilon_{310} = 2.0 \times 10^4 \text{ L} \cdot \text{mol}^{-1} \cdot \text{cm}^{-1}$ for a model 15-bp duplex containing three U^{OH}-U^{OH} pairs (Ref. 28)). Since the conformational changes of the DNAzyme may affect the spectra, a more detailed mechanistic study is required for a quantitative discussion, which will be reported elsewhere in the near

future. We have added these comments to the figure legend of Supplementary Fig. S11 as shown below.

“Supplementary Fig. S11 | UV absorption spectra of U^{OH}-DNAzyme in the absence and presence of Gd^{III} ions. [U^{OH}-DNAzyme] = 20 μM, [GdCl₃] = 0 or 60 μM (3 equiv) in 10 mM HEPES (pH 7.0), 100 mM NaCl, *l* = 0.1 cm, rt. The samples were annealed before the measurement. From the increase in absorbance at 310 nm ($\Delta\epsilon_{310} = 1.5 \times 10^4 \text{ L}\cdot\text{mol}^{-1}\cdot\text{cm}^{-1}$), it was roughly estimated that more than 70% of the U^{OH} bases formed the U^{OH}-Gd^{III}-U^{OH} base pairs ($\Delta\epsilon_{310} = 2.0 \times 10^4 \text{ L}\cdot\text{mol}^{-1}\cdot\text{cm}^{-1}$ for a 15-bp duplex containing three U^{OH}-Gd^{III}-U^{OH} pairs²⁸).”

Reviewer #2

Comment 1. In summary, the authors present a highly innovative and important contribution to the field of dynamic DNA nanotechnology. The use of Gd(III)-mediated base pairs certainly expands the scope of DNA nanotechnology. The manuscript is of interest to readers from various chemical disciplines. I consider it highly important and recommend its acceptance after minor revisions as follows.

Answer 1. We sincerely appreciate this reviewer’s positive assessment of our work.

Comment 2. The authors should briefly state why they chose to use four consecutive UOH residues in all their oligonucleotides. Why not fewer (or more)?

Answer 2. We appreciate this invaluable comment. Our previous studies (Ref. 28) suggested that the Gd^{III}-dependent duplex stabilization is achieved by introducing three or more consecutive U^{OH}-U^{OH} mismatch pairs. We have originally designed 15-mer DNA strands with three U^{OH} bases in addition to the 16-mer strands with four U^{OH} bases. Melting experiments in the absence and presence of Gd^{III} ions revealed that the degree of duplex (de)stabilization (ΔT_m) was slightly greater when four U^{OH} bases were incorporated. Incorporation of more U^{OH} bases was thought to result in undesirable intramolecular metal complexation. Therefore, DNA strands **1** and **2** containing four consecutive U^{OH} bases were chosen for the SDR experiments. These discussions are briefly described in the main text with an additional Supplementary Figure S3. The comparison of the T_m values has been additionally shown as Supplementary Figure S4.

“Prior to the SDR experiments, metal-mediated regulation of DNA hybridization was first studied using oligonucleotides containing four U^{OH} bases. Three 16-mer DNA strands were designed so that strand **1 can hybridize with strand **2** via U^{OH}-Gd^{III}-U^{OH}**

base pairing and with strand **3** via $\text{U}^{\text{OH}}\text{-A}$ base pairing at their termini (Fig. 2a). (snip) As a result, duplex **1·2** containing $\text{U}^{\text{OH}}\text{-Gd}^{\text{III}}\text{-U}^{\text{OH}}$ base pairs was found to be much more stable than duplex **1·3** containing $\text{U}^{\text{OH}}\text{-A}$ base pairs upon addition of Gd^{III} (Fig. 2b). These results indicate that the addition of Gd^{III} ions reversed the order of the stability of the duplexes. The same trend was observed for DNA strands containing three U^{OH} bases (Supplementary Fig. S3). Addition of Gd^{III} ions (3 equiv) stabilized duplex **1'·2'** with three $\text{U}^{\text{OH}}\text{-U}^{\text{OH}}$ pairs ($\Delta T_m = +25.2$ °C) and destabilized duplex **1'·3'** with three $\text{U}^{\text{OH}}\text{-A}$ pairs ($\Delta T_m = -4.4$ °C). The degree of duplex (de)stabilization (ΔT_m) was slightly greater when four U^{OH} bases were incorporated (Supplementary Fig. S4). Incorporation of more than four U^{OH} bases was thought to cause undesirable intramolecular metal complexation. Therefore, DNA strands **1** and **2** containing four consecutive U^{OH} bases were employed in the SDR experiments.”

“Supplementary Fig. S3 | Gd^{III} -mediated stabilization and destabilization of DNA duplexes containing three $\text{U}^{\text{OH}}\text{-U}^{\text{OH}}$ or $\text{U}^{\text{OH}}\text{-A}$ base pairs. **a** Melting curves of a DNA duplex containing three $\text{U}^{\text{OH}}\text{-U}^{\text{OH}}$ base pairs (**1'·2'**) in the presence of different concentrations of Gd^{III} ions. 0.2 °C/min. **b** UV absorption spectra of duplex **1'·2'** in the presence of different concentrations of Gd^{III} ions. 5 °C, $l = 1$ cm. The spectra varied almost linearly in the range $[\text{Gd}^{\text{III}}]/[\text{duplex}] = 0$ to 3 and remained almost unchanged in the presence of excess Gd^{III} ions, suggesting that three Gd^{III} ions bind to the three $\text{U}^{\text{OH}}\text{-U}^{\text{OH}}$ base pairs. **c** Melting curves of a DNA duplex containing three $\text{U}^{\text{OH}}\text{-A}$ base pairs (**1'·3'**) in the presence of different concentrations of Gd^{III} ions. $[\text{Gd}^{\text{III}}]/[\text{duplex}] = 0, 1, 2, 3$ (solid lines), 4 (broken line), and 6 (dotted line), in 10 mM HEPES buffer (pH 8.0), 100 mM NaCl.”

“**Supplementary Fig. S4 | Melting temperatures (T_m) of duplexes with $U^{OH}-U^{OH}$ base pairs and duplexes with $U^{OH}-A$ base pairs in the absence and presence of Gd^{III} ions.** **a** Duplex 1'·2' (with three $U^{OH}-U^{OH}$ pairs) and duplex 1'·3' (with three $U^{OH}-A$ pairs). **b** Duplex 1·2 (with four $U^{OH}-U^{OH}$ pairs) and duplex 1·3 (with four $U^{OH}-A$ pairs) (reprinted from Fig. 2b). [DNA strand] = 2.0 μ M each, [$GdCl_3$] = 0, 6.0, or 8.0 μ M (1 equiv for $U^{OH}-U^{OH}$ pair) in 10 mM HEPES (pH 8.0), 100 mM NaCl. $N = 3$. Error bars indicate standard errors.”

Comment 3. The mass spectrum shown in Figure S1c shows peaks for duplexes bearing three and four Gd(III) ions with similar intensities. Even though this is not relevant for the SDRs, it would be interesting to know whether indeed all four binding sites are occupied by Gd(III) or whether one regularly remains unoccupied. Do the authors have any indication in this respect?

Answer 3. We thank this reviewer for the important question. The UV titration experiment (Fig. S1b) showed that the spectra varied almost linearly in the range [Gd^{III}]/[duplex] = 0 to 4 and remained almost unchanged in the presence of excess Gd^{III} ions. This result suggests that all the U^{OH} bases bind to the Gd^{III} ions. We speculate that the outermost Gd^{III} ion dissociates easily during the mass measurement, in a manner analogous to the fraying of the terminal base pairs of natural duplexes. The use of NH_4OAc as a volatile buffer may facilitate the dissociation of Gd^{III} ions because NH_3 and AcO^- act as competing ligands. These explanations are added in the legend of Supplementary Fig. S1.

“**Supplementary Fig. S1 | Gd^{III} -mediated stabilization of DNA duplex containing $U^{OH}-U^{OH}$ base pairs.** (snip) **b** UV absorption spectra of duplex 1·2 in the presence of different concentrations of Gd^{III} ions. [Gd^{III}]/[duplex] = 0, 1, 2, 3, 4 (solid lines), 5, and 6 (broken lines). In 10 mM HEPES buffer (pH 8.0), 100 mM NaCl. 5 °C, $l = 1$ cm. The spectra varied almost linearly in the range [Gd^{III}]/[duplex] = 0 to 4 and remained almost unchanged in the presence of excess Gd^{III} ions, suggesting that four Gd^{III} ions bind to the four $U^{OH}-U^{OH}$ base pairs. **c** ESI-TOF mass spectrum of duplex 1·2 in the presence of Gd^{III} ions (4 equiv). [duplex] = 40 μ M, [Gd^{III}]/[duplex] = 4.0 in 20 mM NH_4OAc buffer (pH 7.0). Negative mode. The existence of the trinuclear complex ($1\cdot2\cdot Gd^{III}_3$) is likely due to the dissociation of the outermost Gd^{III} ion during the measurement.”

Comment 4. The difference between Figures 2c and S3 should be stated in the text (presumably visualization via FAM fluorescence to show only constructs containing oligonucleotide 3 and via SYBR gold staining to show all constructs?). Moreover, the identity of the “markers” should be stated in the figure legend. Are these molecular weight markers?

Answer 4. We thank the reviewer for the important suggestions. As pointed by the reviewer, we have provided additional explanation in the main text on how to visualize the gels in Fig. 2c (FAM fluorescence) and Fig. S5 (renumbered) (SYBR Gold staining). Authentic samples (strand 3 and pre-annealed duplex 1·3) were used as “markers” in the PAGE analysis. The marker identification has also been added to the figure legends.

“Next, an equimolar mixture of strands 1, 2, and 3 (labeled with a fluorophore FAM) was annealed in the absence and presence of Gd^{III} ions to see which duplex (1·2 or 1·3) is preferentially formed. Native polyacrylamide gel electrophoresis (PAGE) analysis confirmed products containing strand 3 by visualization with FAM fluorescence (Fig. 2c) and all products by further staining (Supplementary Fig. S5). The results revealed that duplex 1·3 was mainly formed (81%) under the Gd^{III}-free condition.”

“**Fig. 2 | Metal-dependent regulation of the hybridization behavior of a DNA strand containing U^{OH} bases.** (snip) c Native PAGE analysis of an equimolar mixture of strands 1, 2, and 3 in the absence and presence of Gd^{III} ions (4 equiv). The samples were annealed prior to the analysis. The strand 3 was labeled with FAM for detection. The authentic samples (strand 3 and pre-annealed duplex 1·3) were employed as the markers.”

“**Supplementary Fig. S5 | Native PAGE analysis of an equimolar mixture of strands 1, 2, and 3 in the absence and presence of Gd^{III} ions (4 equiv).** [DNA duplex] = 2.0 μM, [GdCl₃] = 0 or 8.0 μM (1 equiv per U^{OH}-U^{OH} base pair) in 10 mM HEPES (pH 8.0), 100 mM NaCl. The sample was annealed prior to the analysis. Detected after SYBR Gold staining. The authentic samples (strand 3 and pre-annealed duplex 1·3) were employed as the markers.”

Comment 5. Figure S4 reports data for shortened oligonucleotides 7', 8', and 9'. What was the reason for not using full-length oligonucleotides 7, 8, and 9?

Answer 5. We thank the reviewer for the important comments. The oligonucleotides used in the SDR experiments (7, 8, and 9) are 26-nt strands, which are too long for melting analysis due to their high thermal stability. Therefore, we decided to use shorter DNA strands (7', 8' and 9') to accurately estimate the Gd^{III}-dependent stabilization and destabilization of the duplexes. We have added this explanation to the legend of Fig, S6 (renumbered).

“**Supplementary Fig. S6 | Thermal stability of model DNA duplexes containing four U^{OH}-U^{OH} or U^{OH}-A base pairs with a terminal G-C base pair.** ... c Melting temperatures (T_m) of duplexes 7'·8' (with U^{OH}-U^{OH} base pairs) and 7'·9' (with U^{OH}-A

base pairs) in the absence and presence of Gd^{III} ions (4 equiv). Shorter DNA strands 7', 8', and 9' (17-nt) were utilized because oligonucleotides 7, 8, and 9 (26-nt) used in the strand displacement reactions were too long for the melting analysis. 7': 5'-CAC ATT GTT GTA U^{OH}U^{OH}U^{OH}U^{OH}C-3', 8': 5'-GU^{OH}U^{OH}U^{OH}U^{OH}T ACA ACA ATG TG-3', 9': 5'- GAA AAT ACA ACA ATG TG -3'."

Comment 6. Figure 3b could be improved by including an inset with an expanded section for the first hour of reaction.

Answer 6. We thank the reviewer for this suggestion. As the reviewer suggested, we have added the inset to Fig. 3b to show the expanded section of the time course as shown below.

Comment 7. Line 144: The authors state that “The slow rate of the EDTA-induced SDRs could be improved by increasing the number of UOH bases or by adding terminal bases that facilitate the binding of the incoming strand.”. The latter statement is not in agreement with the Figure 3c, which shows that the presence of one terminal base in fact reduced the rate.

Answer 7. We thank the reviewer for this valuable comment. What we wanted to state is that terminal bases can be added only to the toehold and the incoming strand (not to the incumbent strand) to facilitate SDR. Therefore, the description was revised as follows.

“The slow rate of the EDTA-induced SDRs could be improved by increasing the number of U^{OH} bases or by adding terminal bases only to the toehold and the incoming strand.”

Comment 8. Figure 4c shows a band referred to as “dimer*”. This dimer is neither explained in the text nor in the figure legend.

Answer 8. We appreciate the reviewer for this important comment. The band referred to as “dimer*” is a dimer structure in which two DNA tweezers are connected by binding of strand **d**. Such a dimer structure was also observed in the original DNA tweezers reported by Yurke et al. (Ref. 31). We have added this explanation to the figure legend as follows.

“**Fig. 4 | Metal-mediated operation of DNA tweezers with U^{OH} bases.** (snip) Native PAGE analysis of the structures of DNA tweezers in the absence and presence of Gd^{III} ions. Strand **b** was labeled with FAM for detection. The DNA tweezers without strand **d** (open state) and with a closing strand containing T in place of U^{OH} (closed state) were used as the controls. 15% gel at 20 °C. (*) A dimeric structure³¹ in which two DNA tweezers are connected by binding of strand **d**.”

Reviewer #3

Comment 1. This is an interesting and exhaustive study on DNA nanodevices based on the ability of 5-hydroxyuracil to form a canonical Watson—Crick base pair with adenine as well as a Gd(III)-mediated homo base pair. The manuscript is very well written, the experiments elegantly designed and carefully executed and the conclusions fully consistent with the data obtained.

Answer 1. We sincerely thank the reviewer for the positive assessment of our work.

Comment 2. However, the authors should be more forthcoming regarding comparison of the present study with previous ones from the same group. For example, the U^{OH}-Gd(III)-U^{OH} base pair has been reported previously (references 28—30) and the conceptual novelty of the metal-dependent allosteric DNazymes is also questionable in light of references 46—49. If the DNzyme presented in this paper shows some improvement over those described in references 46—49, it should be explained more clearly. I leave it to the Editor to decide whether the novelty of the results merits publication in Nature Communications or whether a less demanding journal, such as Scientific Reports, would be more appropriate.

Answer 2. We thank the reviewer for the important comments. In our previous study (Ref. 47), we have developed a Cu^{II}-responsive DNzyme by incorporating an H—Cu^{II}—H base pair into the same parent DNzyme (NaA43), resulting in a Cu^{II}-dependent increase in the activity as indicated by $k_{\text{Cu}^+}/k_{\text{Cu}^-} = 5.9$. The U^{OH}-modified DNzyme developed in this study was more efficiently activated by the addition of Gd^{III} ions ($k_{\text{Gd}^+}/k_{\text{Gd}^-} = 14.4$). This may be due to a novel design strategy based on the bifacial nature of the U^{OH} bases, which allowed the stable formation of the inactive secondary structure (with U^{OH}—A base pairs) and the active structure (with U^{OH}—Gd^{III}—U^{OH} pairs) in the absence and presence of Gd^{III} ions, respectively. This design concept for U^{OH}-DNzyme is conceptually new. According to the reviewer’s suggestion, these improvements are described in more detail as follows.

“It can be concluded that the activity of U^{OH} -DNAzyme is allosterically enhanced by the formation of $\text{U}^{\text{OH}}\text{-Gd}^{\text{III}}\text{-U}^{\text{OH}}$ base pairs, stabilizing the catalytically active structure. It should also be noted that the U^{OH} -DNAzyme showed more efficient switching capability ($k_{\text{Gd}^+}/k_{\text{Gd}^-} = 14.4$) than the Cu^{II} -responsive DNAzyme ($k_{\text{Cu}^+}/k_{\text{Cu}^-} = 5.9$)⁴⁷ developed by incorporating an $\text{H-Cu}^{\text{II}}\text{-H}$ base pair into the same parent DNAzyme. This is thought to be due to a new design strategy based on the bifacial nature of the U^{OH} bases, whereby the inactive secondary structure (with $\text{U}^{\text{OH}}\text{-A}$ base pairs) and the active structure (with $\text{U}^{\text{OH}}\text{-Gd}^{\text{III}}\text{-U}^{\text{OH}}$ pairs) are formed stably in the absence and presence of Gd^{III} ions, respectively.”

Reviewer #4

Comment 1. The idea of metal ion mediated strand displacement to control nucleic acid molecular devices is interesting and the authors have demonstrated the working principles of this method. Overall it is a good study. Here are a few questions that need to be addressed before its publication:

Answer 1. We sincerely appreciate the positive feedback from the reviewer.

Comment 2. How versatile this methods is, can this be extended to a wide range of metal ions. Why do people want to use these metals to drive the strand displacement, what is the biological function if any. Isothermal DSD is definitely interesting.

Answer 2. We thank this reviewer for the important comments. Our concept of metal-mediated base pair switching can be extended to other metal ions by designing other bifacial nucleobases with a different metal affinity. Various types of bifacial nucleobases can be developed by changing the coordinating functionality at the 5-position or by modifying the cytosine (C) nucleobases. In fact, we have developed other types of bifacial nucleobases, 5-carboxyuracil (**caU**; *J. Am. Chem. Soc.*, **2020**) and *N,N*-dicarboxymethyl-5-aminouracil (**dcaU**; *Chem. Sci.*, **2023**), both of which were found to form more stabilizing metal-mediated base pairs (**caU**- Cu^{II} -**caU** and **dcaU**- Gd^{III} -**dcaU**) as well as hydrogen-bonded base pairs (**caU**-A and **dcaU**-A) (Fig. R1).

Fig. R1 | Other bifacial nucleobases that form both hydrogen-bonded and metal-mediated base pairs. a 5-carboxyuracil (**caU**) nucleobase. **b** *N,N*-dicarboxymethyl-5-aminouracil (**dcaU**) nucleobase.

We have added these comments to the Discussion part as shown below, and additionally cited our recent paper reporting the synthesis of **dcaU** nucleobase. We are also working on the development of various types of bifacial nucleobases, which will be reported elsewhere in the near future.

“Metal-mediated base-pair switching can be also possible with other 5-substituted pyrimidine bases such as 5-carboxyuracil (**caU**)⁵⁸ and *N,N*-dicarboxymethyl-5-aminouracil (**dcaU**)⁵⁹, both of which were found to form more stabilizing metal-mediated base pairs (**caU**-Cu^{II}-**caU** and **dcaU**-Gd^{III}-**dcaU**) as well as hydrogen-bonded base pairs (**caU**-A and **dcaU**-A). We believe that this method can be extended to other metal ions by designing bifacial nucleobases with different coordination functionality at the 5-position or by modifying cytosine bases. Thus, the principle of metal-mediated base-pair switching appears to be a versatile approach that extends the scope of dynamic DNA nanotechnology.”

“59. Mori, K., Takezawa, Y. & Shionoya, M. Metal-dependent base pairing of bifacial iminodiacetic acid-modified uracil bases for switching DNA hybridization partner. *Chem. Sci.* **14**, 1082–1088 (2023).”

The advantage of using metal ions as a trigger for SDRs were summarized in the Discussion part. According to the reviewer’s comments, we have additionally described the usefulness of the metal-triggered SDRs and future applications using biologically important metal ions, which is one of our next goals. We think biologically relevant Cu^{II} ions will be available by using **caU** nucleobases and such studies are currently underway.

“The use of metal ions as external stimuli in dynamic DNA nanotechnology has the following characteristics. (snip) (ii) Metal coordination is fundamentally reversible,

allowing repeated manipulation of metal-DNA systems, as demonstrated with the DNA tweezers in this study. Reversible SDRs could be useful in the developing advanced DNA computing circuits. (snip) (iv) SDRs triggered by biologically relevant metal ions have a wide range of biological applications.”

Comment 3. in Figure 3 in the gel showing the reversible switching, there appears to be less than 100% of switching, what is causing this? Will this negatively impact any downstream DSD if more complicated devices are designed?

Answer 3. We thank this reviewer for providing these critical comments. The Gd^{III}-mediated control of hybridization of U^{OH}-modified strands (Fig. 2) essentially depends on the balance of thermodynamic stability of the two duplexes, i.e., one with U^{OH}-A base pairs and the other with U^{OH}-U^{OH} pairs. In the case of the Gd^{III}-triggered isothermal SDRs (Fig. 3), the lower efficiency may be due to the formation of undesirable complexes such as homo-dimers of the U^{OH}-containing invading strand through U^{OH}-Gd^{III}-U^{OH} base pairing (Supplementary Fig. S8). Such low efficiency is not a serious problem because the output signal can be amplified using appropriate amplification circuits or catalytic DNAzyme reactions.

We believe that SDRs using U^{OH}-containing strands have additional advantages. Metal-triggered SDRs based on T-Hg^{II}-T base pairing have been previously reported (Ref. 37). Hg^{II} ions may bind to other T bases in undesirable positions, which may adversely affect downstream SDRs. On the other hand, the Gd^{III} ions used in this study interact very little with natural nucleobases. Therefore, Gd^{III}-triggered SDRs using unnatural U^{OH} nucleobases would have a significant advantage, especially when integrated into cascade SDRs such as computing circuits. These viewpoints were further explained as follows.

“Note that the SDRs developed in this study are reversible even though there are no exposed toehold regions in the strands. Since Gd^{III} ions rarely interact with natural nucleobases, the Gd^{III}-triggered SDRs using unnatural U^{OH} nucleobases would have significant advantages, especially when integrated into cascade SDRs such as DNA computing circuits. Thus, SDRs based on the base pair switching between U^{OH}-A and U^{OH}-Gd^{III}-U^{OH} may provide a new strategy for designing wide variety of stimuli-responsive DNA molecular systems.”

Comment 4. The system can be more systematic by including studies of length/sequence/structure dependence. It would improve the work by include the use of metal mediated strand displacement in reconfiguration of DNA nanostructure (1D/2D systems).

Answer 4. We thank the reviewer’s deep insight and valuable suggestions. According to the reviewer’s comment and the Editor’s additional suggestions, we have conducted additional

experiments to apply the metal-mediated base-pair switching to reconfiguration of DNA nanoarchitectures. We have tried metal-triggered assembly of DNA nanotubes based on the strand displacement reaction (SDR) of U^{OH} -containing strands. Although Gd^{III} -induced aggregation was observed under some conditions, it needs further investigations to obtain reliable results (detailed data are disclosed for the reviewers only).

These attempts suggested that for more practical applications in DNA nanotechnology, where DNA concentrations are typically in the nM to sub- μM range, metal-mediated SDR needs to be performed at lower concentrations. Introducing more U^{OH} bases and optimizing the sequence may improve results, but for metal-mediated SDR at low concentrations, the use of other types of bifacial 5-modified uracil bases may be more effective. We have already developed other bifacial nucleobases that can form both hydrogen-bonded and metal-mediated base pairs, such as *N,N*-dicarboxymethyl-5-aminouracil (**dcaU**; *Chem. Sci.*, **2023**) and 5-carboxyuracil bases (**caU**; *J. Am. Chem. Soc.*, **2020**) (Fig. R1). The tetradentate **dcaU** base shows a better affinity to Gd^{III} ions than the bidentate U^{OH} bases and is capable of stabilizing DNA duplexes by forming only a single **dcaU**– Gd^{III} –**dcaU** base pair ($\Delta T_{\text{m}} = +16.1$ °C). The **caU** bases form Cu^{II} -mediated **caU**– Cu^{II} –**caU** base pairs, which stabilize DNA duplexes more efficiently than U^{OH} – Gd^{III} – U^{OH} bases ($\Delta T_{\text{m}} = +30.7$ °C by three consecutive **caU**– Cu^{II} –**caU**). Accordingly, we have decided to employ **dcaU** or **caU** bases for metal-triggered DNA nano-assembly based on the metal-mediated base-pair switching. Because of the potential interaction of Gd^{III} ions with phosphate groups, we believe that other metal ions, such as Cu^{II} , are more suitable for applications in more complex systems consisting of many DNA strands. Metal-mediated control of DNA nanoarchitectures using **dcaU** or **caU** nucleobases will be reported elsewhere in the near future.

We believe that the concept of the metal-mediated base-pair switching is applicable to other 5-modified pyrimidine nucleobases and, therefore, this study provides a new approach to expand the scope of dynamic DNA nanotechnology. To clearly demonstrate the importance and the novelty of this study, we have additionally discussed the generality of the metal-mediated base-pair switching in the conclusion, using the examples of **dcaU** and **caU** nucleobases, as shown below.

“It should also be mentioned that metal-mediated SDR at lower concentrations is required for the practical application in DNA nanotechnology. Because of the potential interaction of Gd^{III} ions with phosphate groups, the use of other metal ions that rarely interact with natural DNA may be more appropriate for applications in complex systems consisting of many DNA strands. Metal-mediated base-pair switching can be also possible with other 5-substituted pyrimidine bases such as 5-carboxyuracil (**caU**)⁵⁸ and *N,N*-dicarboxymethyl-5-aminouracil (**dcaU**)⁵⁹, both of which were found to form more stabilizing metal-mediated base pairs (**caU**– Cu^{II} –**caU** and **dcaU**– Gd^{III} –**dcaU**) as well as hydrogen-bonded base pairs (**caU**–A and **dcaU**–A). We believe that this method can be

extended to other metal ions by designing bifacial nucleobases with different coordination functionality at the 5-position or by modifying cytosine bases. Thus, the principle of metal-mediated base-pair switching appears to be a versatile approach that extends the scope of dynamic DNA nanotechnology. The application of this bifacial nucleobases to the metal-dependent reconfiguration of DNA nanostructures is currently under investigation in our laboratory.”

Additional corrections

- (1) The sequences and MS data of the DNA strands containing three U^{OH} bases (**ODN1'** and **ODN2'**) were added to Supplementary Tables S1 and S2.
- (2) The MS data of **ODN1** in Supplementary Table S2 have been corrected because incorrect values were mistakenly shown.
- (3) A related paper that we have published recently are additionally cited.
“50. Rajasree, S. C., Takezawa, Y. & Shionoya, M. Cu(II)-mediated stabilisation of DNA duplexes bearing consecutive ethenoadenine lesions and its application to a metal-responsive DNAzyme. *Chem. Commun.* **59**, 1006–1009 (2023).”

REVIEWERS' COMMENTS

Reviewer #1 (Remarks to the Author):

The authors have addressed all the comments in the original review well and I would be happy with acceptance at this stage.

Reviewer #2 (Remarks to the Author):

The authors provide convincing responses to all reviewers' comments. Additional experiments were performed and their results included in the revised manuscript. In my opinion, this very good manuscript was improved even further during the revision. I recommend acceptance without the necessity of any further changes.

Reviewer #4 (Remarks to the Author):

The authors have addressed my questions. I recommend it for publication.